# Bi-level Physics-Informed Neural Networks for PDE Constrained Optimization using Broyden's Hypergradients

**Zhongkai Hao**[1,2,3]**, Chengyang Ying**[1]**, Hang Su**[1,4]**, Jun Zhu**[1,3,4]***,Jian Song**[2]**, Ze Cheng**[5]

[1]Dept. of Comp. Sci. & Tech., Institute for AI, BNRist Center, THBI Lab,
Tsinghua-Bosch Joint Center for ML, Tsinghua University
[2]Dept. of EE, Tsinghua University,
[3] RealAI, [4]Pazhou Lab, Guangzhou, 510330, China,[5]Bosch China Investment Ltd
`{hzj21, ycy21}@mail.tsinghua.edu.cn,`
`{dcszj, suhangss, jsong}@tsinghua.edu.cn, ze.cheng@cn.bosch.com`

## Abstract

Deep learning based approaches like Physics-informed neural networks (PINNs) and DeepONets have shown promise on solving PDE constrained optimization (PDECO) problems. However, existing methods are insufficient to handle those PDE constraints that have a complicated or nonlinear dependency on optimization targets. In this paper, we present a novel bi-level optimization framework to resolve the challenge by decoupling the optimization of the targets and constraints. For the inner loop optimization, we adopt PINNs to solve the PDE constraints only. For the outer loop, we design a novel method by using Broyden's method based on the Implicit Function Theorem (IFT), which is efficient and accurate for approximating hypergradients. We further present theoretical explanations and error analysis of the hypergradients computation. Extensive experiments on multiple large-scale and nonlinear PDE constrained optimization problems demonstrate that our method achieves state-of-the-art results compared with strong baselines.

## 1 Introduction

PDE constrained optimization (PDECO) aims at optimizing the performance of a physical system constrained by partial differential equations (PDEs) with desired properties. It is a fundamental task in numerous areas of science (Chakrabarty & Hanson, 2005; Ng & Dubljevic, 2012) and engineering (Hicks & Henne, 1978; Chen et al., 2009), with a wide range of important applications including image denoising in computer vision (De los Reyes & Schönlieb, 2013), design of aircraft wings in aerodynamics (Hicks & Henne, 1978), and drug delivery (Chakrabarty & Hanson, 2005) in biology etc. These problem have numerous inherent challenges due to the diversity and complexity of physical constraints and practical problems.

Traditional numerical methods like adjoint methods (Herzog & Kunisch, 2010) based on finite element methods (FEMs) (Zienkiewicz et al., 2005) have been studied for decades. They could be divided into continuous and discretized adjoint methods (Mitusch et al., 2019). The former one requires complex handcraft derivation of adjoint PDEs and the latter one is more flexible and more frequently used. However, the computational cost of FEMs grows quadratically to cubically (Xue et al., 2020) w.r.t mesh sizes. Thus compared with other constrained optimization problems, it is much more expensive or even intractable to solve high dimensional PDECO problems with a large search space or mesh size.

To mitigate this problem, neural network methods like DeepONet (Lu et al., 2019) have been proposed as surrogate models of FEMs recently. DeepONet learns a mapping from control (decision) variables to solutions of PDEs and further replaces PDE constraints with the operator network. But these methods require pretraining a large operator network which is non-trivial and inefficient. Moreover,

---

*Corresponding author.

its performance may deteriorate if the optimal solution is out of the training distribution (Lanthaler et al., 2022). Another approach of neural methods (Lu et al., 2021; Mowlavi & Nabi, 2021) proposes to use a single PINN (Raissi et al., 2019) to solve the PDECO problem instead of pretraining an operator network. It uses the method of Lagrangian multipliers to treat the PDE constraints as regularization terms, and thus optimize the objective and PDE loss simultaneously. However, such methods introduce a trade-off between optimization targets and regularization terms (i.e., PDE losses) which is crucial for the performance (Nandwani et al., 2019). It is generally non-trivial to set proper weights for balancing these terms due to the lack of theoretical guidance. Existing heuristic approaches for selecting the weights may usually yield an unstable training process. Therefore, it is imperative to develop an effective strategy to handle PDE constraints for solving PDECO problems.

To address the aforementioned challenges, we propose a novel bi-level optimization framework named Bi-level Physics-informed Neural networks with Broyden's hypergradients (BPN) for solving PDE constrained optimization problems. Specifically, we first present a bi-level formulation of the PDECO problems, which decouples the optimization of the targets and PDE constraints, thereby naturally addressing the challenge of loss balancing in regularization based methods. To solve the bi-level optimization problem, we develop an iterative method that optimizes PINNs with PDE constraints in the inner loop while optimizes the control variables for objective functions in the outer loop using hypergradients. In general, it is nontrivial to compute hypergradients in bi-level optimization for control variables especially if the inner loop optimization is complicated (Lorraine et al., 2020). To address this issue, we further propose a novel strategy based on implicit differentiation using Broyden's method which is a scalable and efficient quasi-Newton method in practice (Kelley, 1995; Bai et al., 2020). We then theoretically prove an upper bound for the approximation of hypergradients under mild assumptions. Extensive experiments on several benchmark PDE constrained optimization problems show that our method is more effective and efficient compared with the alternative methods.

We summarize our contributions as follows:

- To the best of our knowledge, it is the first attempt that solves general PDECO problems based on deep learning using a bi-level optimization framework that enjoys scalability and theoretical guarantee.
- We propose a novel and efficient method for hypergradients computation using Broyden's method to solve the bi-level optimization.
- We conduct extensive experiments and achieve state-of-the-art results among deep learning methods on several challenging PDECO problems with complex geometry or non-linear Naiver-Stokes equations.

## 2 RELATED WORK

**Neural Networks Approaches for PDE Constrained Optimization.** Surrogate modeling is an important class of methods for PDE constrained optimization (Queipo et al., 2005). Physics-informed neural networks (PINNs) are powerful and flexible surrogates to represent the solutions of PDEs (Raissi et al., 2019). hPINN (Lu et al., 2021) treats PDE constraints as regularization terms and optimizes the control variables and states simultaneously. It uses the penalty method and the Lagrangian method to adjust the weights of multipliers. (Mowlavi & Nabi, 2021) also adopts the same formulation but uses a line search to find the largest weight when the PDE error is within a certain range. The key limitation of these approaches is that heuristically choosing methods for tuning weights of multipliers might be sub-optimal and unstable. Another class of methods train an operator network from control variables to solutions of PDEs or objective functions. Several works (Xue et al., 2020; Sun et al., 2021; Beatson et al., 2020) use mesh-based methods and predict states on all mesh points from control variables at the same time. PI-DeepONet (Wang et al., 2021a;c) adopts the architecture of DeepONet (Lu et al., 2019) and trains the network using physics-informed losses (PDE losses). However, they produce unsatisfactory results if the optimal solution is out of the distribution (Lanthaler et al., 2022).

**Bi-level Optimization in Machine Learning.** Bi-level optimization is widely used in various machine learning tasks, e.g., neural architecture search (Liu et al., 2018), meta learning (Rajeswaran et al., 2019) and hyperparameters optimization (Lorraine et al., 2020; Bao et al., 2021). One of the key challenges is to compute hypergradients with respect to the inner loop optimization (Liu et al., 2021). Some previous works (Maclaurin et al., 2015; Liu et al., 2018) use unrolled optimization or

truncated unrolled optimization which is to differentiate the optimization process. However, this is not scalable if the inner loop optimization is a complicated process. Some other works (Lorraine et al., 2020; Clarke et al., 2021; Rajeswaran et al., 2019) compute the hypergradient based on implicit function theorem. This requires computing of the inverse hessian-vector product (inverse-HVP). In (Lorraine et al., 2020) it proposes to use neumann series to approximate hypergradients. Some works also use the conjugated gradient method (Pedregosa, 2016). The approximation for implicit differentiation is crucial for the accuracy of hypergradients computation (Grazzi et al., 2020).

# 3 METHODOLOGY

## 3.1 PRELIMINARIES

Let $Y, U, V$ be three Banach spaces. The solution fields of PDEs are called state variables, i.e., $y \in Y_{\mathrm{ad}} \subset Y$, and functions or variables we can control are control variables, i.e., $u \in U_{\mathrm{ad}} \subset U$ where $Y_{\mathrm{ad}}, U_{\mathrm{ad}}$ are called admissible spaces, e.g. a subspace parameterized by neural networks or finite element basis. The PDE constrained optimization can be formulated as:

$$\min_{y \in Y_{\mathrm{ad}}, u \in U_{\mathrm{ad}}} \mathcal{J}(y, u), \quad \text{s.t.} \quad e(y, u) = 0, \tag{1}$$

where $\mathcal{J} : Y \times U \to \mathbb{R}$ is the objective function and $e : Y \times U \to V$ are PDE constraints. Usually, the PDE system of $e(y, u) = 0$ contains multiple equations and boundary/initial conditions as

$$\begin{aligned} \mathcal{F}(y, u)(x) &= 0, \quad \forall x \in \Omega \\ \mathcal{B}(y, u)(x) &= 0, \quad \forall x \in \partial\Omega, \end{aligned} \tag{2}$$

where $\mathcal{F} : Y \times U \to (\Omega \to \mathbb{R}^{d_1})$ is the differential operator representing PDEs and $\mathcal{B} : Y \times U \to (\Omega \to \mathbb{R}^{d_2})$ represents boundary/initial conditions. Existing methods based on regularization (e.g. the penalty method) solve the PDECO problem by minimizing the following objective (Lu et al., 2021):

$$\min_{w, \theta} \hat{\mathcal{J}} = \mathcal{J}(y_w, u_\theta) + \int_\Omega |\lambda_1 \cdot \mathcal{F}(y_w, u_\theta)(x)|^2 \mathrm{d}x + \int_{\partial\Omega} |\lambda_2 \cdot \mathcal{B}(y_w, u_\theta)(x)|^2 \mathrm{d}x, \tag{3}$$

where the solutions $y$ and control variables $u$ are respectively parameterized by $w \in \mathbb{R}^m$ and $\theta \in \mathbb{R}^n$ with $w$ being the weights of PINNs. $\lambda_i \in \mathbb{R}^{d_i}$ are hyper-parameters balancing these terms of the optimization targets. One main difficulty is that $\lambda_i$ are hard to set and the results are sensitive to them due to the complex nature of regularization terms (PDE constraints). In general, large $\lambda_i$ makes it difficult to optimize the objective $\mathcal{J}$, while small $\lambda_i$ can result in a nonphysical solution of $y_w$. Besides, the optimal $\lambda_i$ may also vary with the different phases of training.

## 3.2 REFORMULATING PDECO AS BI-LEVEL OPTIMIZATION

To resolve the above challenges of regularization based methods, we first present a new perspective that interprets PDECO as a bi-level optimization problem (Liu et al., 2021), which can facilitate a new solver consequentially. Specifically, we solve the following bi-level optimization problem:

$$\begin{aligned} \min_\theta \quad & \mathcal{J}(w^*, \theta) \tag{4} \\ s.t. \quad & w^* = \arg\min_w \mathcal{E}(w, \theta). \end{aligned}$$

In the outer loop, we only minimize $\mathcal{J}$ with respect to $\theta$ given the optimal value of $w^*$, and we optimize PDE losses using PINNs in the inner loop with the fixed $\theta$. The objective $\mathcal{E}$ of the inner loop sub-problem is:

$$\mathcal{E} = \int_\Omega |\mathcal{F}(y_w, u_\theta)(x)|^2 \mathrm{d}x + \int_{\partial\Omega} |\mathcal{B}(y_w, u_\theta)(x)|^2 \mathrm{d}x. \tag{5}$$

By transforming the problem in Eq. equation 1 into a bi-level optimization problem, the optimization of PDEs' state variables and control variables can be decoupled, which relieves the headache of setting a proper hyper-parameter of $\lambda_i$ in Eq. equation 3.

To solve this bi-level optimization problem, we design inner loops and outer loops that are executed iteratively. As shown in Figure 1, we train PINNs in the inner loop with PDE losses in Eq. equation 5. In the outer loop, we compute the hypergradients based on Implicit Function Differentiation inspired

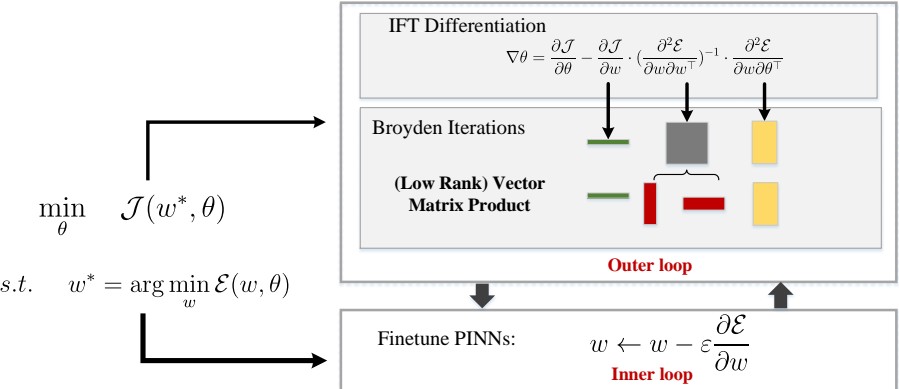

Figure 1: Illustration of our bi-level optimization framework (BPN) for solving PDE constrained optimization problems. In each iteration, we compute hypergradients of control parameters $\theta$ using IFT Differentiation in the outer loop. We calculate inverse vector-Hessian product based on Broyden's method which uses low rank approximation for acceleration. Then in the inner loop, we fine-tune PINNs using PDE losses only.

by Lorraine et al. (2020). Along this line, we need to calculate a highly complex inverse Hessian-Jacobian product. To address this issue, we propose to use Broyden's method which provides an efficient approximation at a superlinear convergence speed (Rodomanov & Nesterov, 2021). In particular, we fine-tune the PINNs using the PDE losses in each iteration of the inner loop. We further compute the gradients of $\mathcal{J}$ with respect to parameters of control variables $\theta$ in the outer loop, which is also recognized as *hypergradients* in bi-level optimization and detailed in the following section.

### 3.3 HYPERGRADIENTS COMPUTATION USING BROYDEN ITERATIONS

The upper-level objective $\mathcal{J}$ depends on the optimal $w^*$ of the lower level optimization, i.e.,

$$\frac{\mathrm{d}\mathcal{J}}{\mathrm{d}\theta} = \frac{\partial \mathcal{J}}{\partial \theta} + \frac{\partial \mathcal{J}}{\partial w^*}\frac{\partial w^*}{\partial \theta}. \tag{6}$$

Thus we need to consider the Jacobian of $w^*$ with respect to $\theta$ when calculating the hypergradients. Since $w^*$ minimizes the lower level problem, we can derive $\frac{\partial w^*}{\partial \theta}$ by applying Cauchy Implicit Function Theorem (Lorraine et al., 2020) as,

**Proposition 1** (Proof in Appendix B.1). *If for some $(w', \theta')$, the lower level optimization is solved, i.e. $\frac{\partial \mathcal{E}}{\partial w}|_{(w',\theta')} = 0$ and $(\frac{\partial^2 \mathcal{E}}{\partial w \partial w^\top})^{-1}$ is invertible, then there exists a function $w^* = w^*(\theta)$ surrounding $(w', \theta')$ s.t. $\frac{\partial \mathcal{E}}{\partial w}|_{(w^*(\theta'),\theta')} = 0$ and we have:*

$$\frac{\partial w^*}{\partial \theta}\bigg|_{\theta'} = -\left[\frac{\partial^2 \mathcal{E}}{\partial w \partial w^\top}\right]^{-1} \cdot \frac{\partial^2 \mathcal{E}}{\partial w \partial \theta^\top}\bigg|_{\left(w^*(\theta'),\theta'\right)}. \tag{7}$$

By Proposition 1, we could compute the hypergradients analytically as

$$\frac{\mathrm{d}\mathcal{J}}{\mathrm{d}\theta} = \frac{\partial \mathcal{J}}{\partial \theta} - \frac{\partial \mathcal{J}}{\partial w^*} \cdot \left(\frac{\partial^2 \mathcal{E}}{\partial w \partial w^\top}\right)^{-1} \cdot \frac{\partial^2 \mathcal{E}}{\partial w \partial \theta^\top}\bigg|_{\left(w^*(\theta),\theta\right)}. \tag{8}$$

However, computing the inverse of Hessian matrix, i.e., $\left(\frac{\partial^2 \mathcal{E}}{\partial w \partial w^\top}\right)^{-1}$, is intractable for parameters of neural networks. To handle this challenge, we can first compute $z^* \triangleq \frac{\partial \mathcal{J}}{\partial w^*} \cdot \left(\frac{\partial^2 \mathcal{E}}{\partial w \partial w^\top}\right)^{-1}$ which is also called the inverse vector-Hessian product. Previous works (Lorraine et al., 2020) use the Neumann series to approximate $z^*$ with linear convergence speed. However, in practice this approach is usually a coarse and imprecise estimation of the hypergradients (Grazzi et al., 2020). Here we employ a more efficient and effective approach to compute the inverse vector-Hessian product which enjoys superlinear convergence speed (Rodomanov & Nesterov, 2021). It equals to finding the root

$z^*$ for the following linear equation as

$$g_w(z) = \frac{\partial}{\partial w} \left( \frac{\partial \mathcal{E}}{\partial w^\top} \cdot z \right) - \frac{\partial \mathcal{J}}{\partial w} \Big|_{w=w^*} = 0. \tag{9}$$

Note that for each evaluation of $g_w(z)$, we only need to compute two Jacobian-vector products with a low computational cost, which does not need to create a giant instance of Hessian matrix. Specifically, we use a low rank Broyden's method (Broyden, 1965; Rodomanov & Nesterov, 2021) to iteratively approximate the solution $z^*$. In each iteration, we first approximate the inverse of $\frac{\partial^2 \mathcal{E}}{\partial w \partial w^\top}$ as

$$\left( \frac{\partial^2 \mathcal{E}}{\partial w \partial w^\top} \right)^{-1} \approx B_i = -I + \sum_{k=1}^{i} u_k v_k^\top. \tag{10}$$

We update $u, v$ and $z$ according to the following rules,

$$z_{i+1} = z_i - \alpha \cdot B_i g_i(z_i) \tag{11}$$

$$u_{i+1} = \frac{\Delta z_{i+1} - B_i \Delta g_{i+1}}{(\Delta z_{i+1})^\top B_i \Delta g_{i+1}} \tag{12}$$

$$v_{i+1} = B_i \Delta z_{i+1} \tag{13}$$

where $\Delta z_{i+1} = z_{i+1} - z_i$, $\Delta g_{i+1} = g_{i+1} - g_i$, and $\alpha$ is the step size (usually set to 1 or using line search). In summary, the inversion of the Hessian could be updated by

$$\left( \frac{\partial^2 \mathcal{E}}{\partial w \partial w^\top} \right)^{-1} \approx B_{i+1} = B_i + \frac{\Delta z_{i+1} - B_i \Delta g_{i+1}}{(\Delta z_{i+1})^\top B_i \Delta g_{i+1}} \Delta z_{i+1}^\top B_i. \tag{14}$$

After $m$ iterations, we use $z_m$ as the approximation of $z^*$. We store $u_i, v_i$ in low rank matrices which is displayed as two red thin matrices in Figure 1. Since we use a low rank approximation of the inverse of Hessian matrix, we no longer need to store the whole matrix and only need to record $u_k$ and $v_k$, where $k = 1 \ldots K$ and $K$ is a tunable parameter depending on the memory limit. We run Broyden iterations until the maximum iteration is reached or the error is within a threshold.

Our method is named Bi-level Physics-informed Neural networks with Broyden's hypergradients (BPN), of which the pseudo code is outlined in Algorithm 1. Given a PDECO problem, we initialize PINNs with random parameters $w_0$ and a guess on control parameters $\theta_0$. First, we train PINNs under initial control $\theta_0$ for $N_w$ epochs as warm up (influence of hyperparameters will be discussed in Appendix C). Then, we compute the hypergradients for $\theta$ using Broyden's method and update it with gradient descent. After that, we fine-tune PINNs under $\theta$ for $N_f$ epochs. These two steps are iteratively executed until convergence.

---

**Algorithm 1** Bi-level Physics-informed Neural networks with Broyden's hypergradients (BPN).

---

**Input:** PINNs $u_w$ with parameters $w_0$, initial guess $\theta_0$, loss functions $\mathcal{E}$ and $\mathcal{J}$, warmup and finetune epochs $N_w, N_f$, learning rates $\epsilon_1, \epsilon_2$ for PINNs and $\theta$ respectively
**Output:** optimal control parameters $\theta$

1: Train $u_w$ under control $\theta_0$ for $N_w$ epochs
2: **for** $i = 1, 2, ..., N_{\text{iter}}$ **do**
3:      Compute hypergradients $\nabla \theta_i$ using Broyden's method in Eq (9) and Eq (14).
4:      $\theta_{i+1} = \theta_i - \epsilon_2 \nabla \theta_i$.
5:      Train $u_w$ under control $\theta_{i+1}$ for $N_f$ epochs.
6:      **if** converged **then**
7:          Break the cycle.
8:      **end if**
9: **end for**

---

## 4 THEORETICAL ANALYSIS AND DISCUSSION

**Error analysis of hypergradients approximation.** We now present analyses of the hypergradients approximation using BPN. Due to the complexity of the problem and computational resources, it is extremely difficult to calculate the hypergradients exactly (Grazzi et al., 2020). Our BPN

using Broyden's method based on Implicit Function Differentiation is also an approximation for hypergradients. Then the approximation error or the convergence rate for hypergradients is one of the key factors that determine the performance. Here we show that the approximation error could be bounded under mild assumptions using Broyden's method. Inspired by the idea of (Grazzi et al., 2020; Ji et al., 2021), we prove that the approximation error could be decomposed into two parts, the former error term is induced by the inner loop optimization and the latter term is caused by the linear solver. Moreover, since our hypergradients are approximated by solving the linear equations using Broyden's method, it enjoys a superlinear convergence rate (Rodomanov & Nesterov, 2021), which is superior compared with other methods such as Neumann Series (Lorraine et al., 2020). Specifically, we state the assumptions to be satisfied as below.

**Assumption 1.** *We denote* $\nabla_1 = \frac{\partial}{\partial w}, \nabla_2 = \frac{\partial}{\partial \theta}, \mathrm{d}_1 = \frac{\mathrm{d}}{\mathrm{d}w}, \mathrm{d}_2 = \frac{\mathrm{d}}{\mathrm{d}\theta}$ *for all* $(w, \theta)$,

- $\nabla_1^2 \mathcal{E}$ *is invertible and* $\|\nabla_1^2 \mathcal{E}\|_2 \geq \mu$.

- $\nabla_1^2 \mathcal{E}$ *and* $\nabla_1 \nabla_2 \mathcal{E}$ *are Lipschitz continuous with constants* $\rho$ *and* $\tau$ *respectively.*

- $\nabla_1 \mathcal{J}, \nabla_2 \mathcal{J}, \nabla_1 \mathcal{E}$ *and* $\nabla_2 \mathcal{E}$ *are Lipschitz continuous with constant* $L$.

- *The inner loop optimization is solved by an iterative algorithm that satisfies* $\|w_t - w\|_2 \leq p_t \|w\|_2$ *and* $p_t < 1, p_t \to 0$ *as* $t \to \infty$ *where* $t$ *is the number of iterations.*

The above assumptions hold for most practical PDECO problems. For the linear equation in Eq. equation 9, we use a $m$ step Broyden's method to solve it. Denote $\kappa = \frac{L}{\mu}$ and $\mathrm{d}_2 J_t, \mathrm{d}_2 J$ are the computed and real hypergradients for $\theta$ respectively, we have the following Theorem.

**Theorem 1** (Proof in Appendix B.2). *If Assumption 1 holds and the linear equation equation 9 is solved with Broyden's method, there exists a constant* $M > 0$, *such that the following inequality holds at iteration* $t$,

$$\|\mathrm{d}_2 J_t - \mathrm{d}_2 J\|_2 \leq \left( L \left( 1 + \kappa + \frac{\rho M}{\mu^2} \right) + \frac{\tau M}{\mu} \right) p_t \|w\|_2 + M\kappa \left( 1 - \frac{1}{m\kappa} \right)^{\frac{m^2}{2}}. \quad (15)$$

The theorem above provides a rigorous theoretical guarantee that the hypergradients are close to real hypergradients when the inner loop optimization, i.e., PINNs' training, could be solved. We observe that the convergence rate relies on the inner loop optimization $p_t$ which is usually locally linear (Jiao et al., 2022) for gradient descent and a superlinear term introduced by Broyden's method (Rodomanov & Nesterov, 2021). The high convergence speed could nearly eliminate this term in a few iterations, so the main error is usually dominated by the first term which is also verified in our experiments.

**Connections to traditional adjoint methods.** Our BPN and adjoint methods (Herzog & Kunisch, 2010) share a similar idea of solving a transposed linear system in Eq. equation 9 to reduce the computational cost. But our bi-level optimization is a more general framework compared with constrained optimization in adjoint methods. It allows more flexible design choices like replacing FEM solvers with PINNs. Additionally, Eq. (9) is different from the adjoint equation (Herzog & Kunisch, 2010). Our Eq (9) solves a system in the parameter space of neural networks but the traditional adjoint equation corresponds to a real field defined on admissible space $U$ or meshes.

## 5 EXPERIMENTS

In this section, we conduct extensive experiments on practical PDE constrained optimization problems to show the effectiveness of our method.

### 5.1 EXPERIMENTAL SETUP AND EVALUATION PROTOCOL

**Benchmark problems.** We choose several classic and challenging PDE constrained optimization problems containing both boundary, domain and time distributed optimization for both linear and non-linear equations. More details of problems are listed in Appendix A.

**(1) Poisson's Equations.** We solve this classic problem on an irregular domain which could be viewed as a prototype of the heat exchanger (Diersch et al., 2011) that is widely used in multiple domains. We denote this problem as Poisson's 2d CG in Table 1.

**(2) Heat Equations.** A time-distributed control problem for two dimensional heat equation similar to (Wang et al., 2021a) (denoted by Heat 2d).

**(3) Burgers Equations.** A time-distributed control problem for burgers equations (Burgers 1d) which are used in nonlinear acoustics and gas dynamics.

**(4)∼(6) Navier-Stokes Equations.** They are highly nonlinear equations characterizing flow of fluids. We solve one shape optimization problem similar to (Wang et al., 2021a) (denoted by NS Shape in Table 1) and two boundary control problems (denoted by NS 2inlets and NS backstep in Table 1) (Mowlavi & Nabi, 2021).

**Baselines.** To demonstrate the effectiveness and superiority of our method, we compare it with several neural methods for PDECO recently proposed.

**(1)∼(2) hPINN (Lu et al., 2021):** It trains PINNs with PDE losses and objective functions jointly with adaptive weights as a regularization term. There are two strategies, i.e. the penalty method and the augmented Lagragian method to update weights and we use hPINN-P and hPINN-A to denote them respectively.

**(3) PINN-LS (Mowlavi & Nabi, 2021):** It also treats the objective function as regularization term with adaptive weights. And it uses line-search rules to find the maximum tolerated weights for the objective function.

**(4) PI-DeepONet (Wang et al., 2021a):** As a two-stage method, it first trains a DeepONet using physics-informed losses and then optimizes the objective using the operator network.

Apart from the methods mentioned above, we also implement several other bi-level optimization algorithms and measure their performance on these PDECO tasks.

**(5) Truncated Unrolled Differentiation(TRMD) (Shaban et al., 2019):** It assumes that the inner loop optimization is solved by several gradient updates and uses reverse mode backpropagation to calculate hypergradients.

**(6) $T1 - T2$ (Luketina et al., 2016):** It computes the hypergradients using Implicit Differentiation Theorem and uses an identity matrix instead of calculating the exact inversion of Hessian matrix.

**(7) Neumann Series (Neumann) (Lorraine et al., 2020):** It proposed to use the Neumann series to iteratively approximates the inverse of Hessian and uses the vector-Jacobian product and the vector-Hessian product to avoid instantiation of Hessian matrices.

Beyond these baselines, we also choose adjoint method (Herzog & Kunisch, 2010) which is a traditional PDECO method as a reference. We run adjoint methods based on high fidelity finite element solvers (Mitusch et al., 2019) to calculate the reference solutions of these tasks. For linear problems, it could be viewed as the ground truth solution. However, it is computationally expensive and sensitive to initial guesses since it solves the whole system in each iteration.

**Hyperparameters and Evaluation Protocol.** We use multi-layer perceptrons (MLP) with width from 64∼128 and depth from $3 \sim 5$ for different problems, and train them using Adam optimizer (Kingma & Ba, 2014) with a learning rate $10^{-3}$. Since the accuracy of PINNs of regularization based methods could not be guaranteed, we resort to finite element methods (FEM) to evaluate the performance. The evaluation metric is the objective function for each problem which is specified in Appendix A. We save and interpolate the control variables and solve the system using FEM and calculate the objective function numerically in every validation epoch. Other details and method-specific hyperparameters are reported in Appendix F. We run experiments on a single 2080 Ti GPU.

## 5.2 MAIN RESULTS

Based on the experimental results for all PDECO tasks in Table 1, we have the following observations. First, our BPN achieves state-of-the-art performance compared with all baselines. It shows that bi-level optimization is an effective and scalable way for large-scale PDECO problems. Second, we observe that our BPN reaches nearly the global optimal result for linear equations (Poisson's and Heat equation). For these equations, the problem is convex and reference values provided by adjoint methods could be viewed as the ground truth. For non-linear problems, the reference values provide locally optimal solutions and our BPN sometimes outperforms adjoint methods. Third, the update strategy of loss weights is critical for regularization based methods like hPINN-P, hPINN-A and

| Objective ($\mathcal{J}$) | Poisson's 2d CG (2d) | Heat 2d (1d) | Burgers 1d (1d) | NS Shape (V) | NS 2inlets (1d) | NS backstep (1d) |
|---|---|---|---|---|---|---|
| Initial guess | 0.400 | 0.142 | 0.0974 | 1.78 | 0.0830 | 0.138 |
| hPINN-P | 0.373 | 0.132 | 0.0941 | – | 0.0732 | 0.0586 |
| hPINN-A | 0.352 | 0.138 | 0.0895 | – | 0.0867 | 0.0696 |
| PINN-LS | 0.239 | 0.112 | 0.0702 | – | 0.0634 | 0.0831 |
| PI-DeepONet | 0.392 | 0.0501 | 0.0710 | **1.27** | 0.0850 | 0.0670 |
| Ours | **0.160** | **0.0379** | **0.0662** | **1.27** | **0.0369** | **0.0365** |
| Reference values | 0.159 | 0.0378 | 0.0634 | 1.27 | 0.0392 | 0.0382 |

Table 1: Main results for performance comparison of different algorithms on several PDECO tasks. Lower score means better performance. We **bold** the best results across all baselines except from the reference values. "–" means that this method cannot solve the problem. "2d"/"1d"/"V" means the control variable is a 2d/1d function or a vector.

| Objective ($\mathcal{J}$) | Poisson's 2d CG | Heat 2d | Burgers 1d | NS Shape | NS 2inlets | NS backstep |
|---|---|---|---|---|---|---|
| Initial guess | 0.400 | 0.142 | 0.0974 | 1.77 | 0.0830 | 0.138 |
| TRMD | 0.224 | 0.0735 | 0.0671 | – | 0.0985 | 0.0391 |
| $T1 - T2$ | 0.425 | 0.0392 | 0.0865 | 1.68 | 0.115 | 0.0879 |
| Neumann | 0.225 | 0.0782 | 0.0703 | 1.31 | 0.0962 | 0.0522 |
| Broyden(Ours) | **0.160** | **0.0379** | **0.0662** | **1.27** | **0.0369** | **0.0365** |

Table 2: Performance comparison for different strategies of computing hypergradients. Lower score means better performance. We **bold** the best results across all methods.

PINN-LS which limits their performance without theoretical guidelines. A possible reason is that the balance between the objective function and PDE losses is sensitive and unstable for complex PDEs.

## 5.3 COMPARISON WITH OTHER BI-LEVEL OPTIMIZATION STRATEGIES

We list the performance of all bi-level optimization methods in Table 2. First, we observe that our method using Broyden's method for hypergradients computation achieves the best results compared with other methods. This is a natural consequence since our method gives the most accurate approximation for the response gradients. Second, we could see that all bi-level optimization methods are effective on solving PDECO problems. Third, the results are better if the hypergradients are more accurate in most cases. For example, Neumann series and TRMD use better approximation for inverse Hessian and they perform better compared with $T1 - T2$.

## 5.4 EXPERIMENTS ON ITERATION EFFICIENCY

To show that our method is computationally efficient, we conduct efficiency experiments by comparing the iterations required. Since PI-DeepONet is a two-stage method and FEMs are not based on gradient descent, we only choose three regularization based methods as baselines. Note that for our BPN we count total inner loop iterations for fairness. We plot values of objective functions $\mathcal{J}$ in Figure 22. We found that our BPN is much more efficient with a nearly linear convergence speed for linear problems. This shows that the hypergradients provided by bi-level optimization are more stable and effective compared with regularization based methods. Besides, we found that PINN-LS is more efficient compared with hPINNs. However, it is a common drawback that regularization based methods are not stable which might take a lot of effort to find suitable loss weights.

## 5.5 FIDELITY OF HYPERGRADIENTS AND ABLATION STUDIES

**Fidelity of hypergradients compared with other methods.** In this experiment, we aim to show how accurate the hypergradients are for these bi-level optimization methods. However, all tasks in our main experiments do not have a closed form solution for response gradients. Here we conduct this experiment on a toy task of one dimensional Poisson's equation with an analytical solution in Appendix A. We first compare our method with several other bi-level optimization methods. We measure the cosine similarity, which is defined as $\frac{x \cdot y}{\|x\| \|y\|}$ for any two vectors $x, y$, between computed hypergradients and analytical hypergradients and the results are shown in Figure 3. Note that the data are collected in the first 75 outer iterations since all methods converge fast on this problem. The left part of the figure shows that Broyden's method gives the most accurate hypergradients with similarity

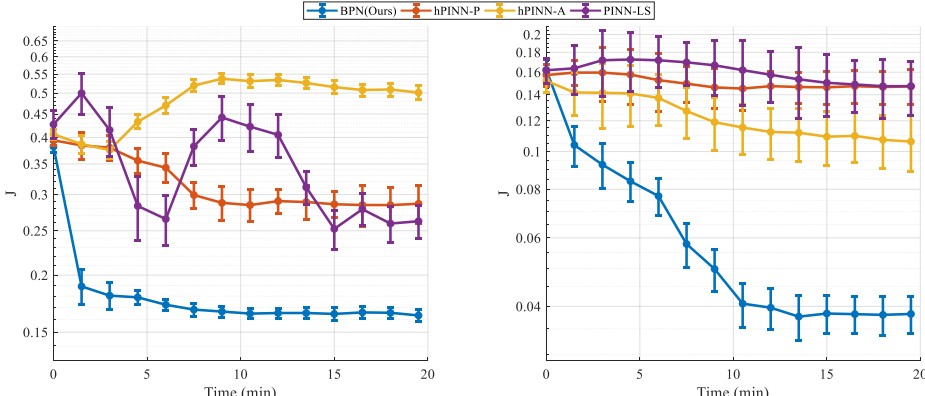

Figure 2: Results of efficiency experiments on Poisson 2d CG (left) and Heat 2d (right) problem.

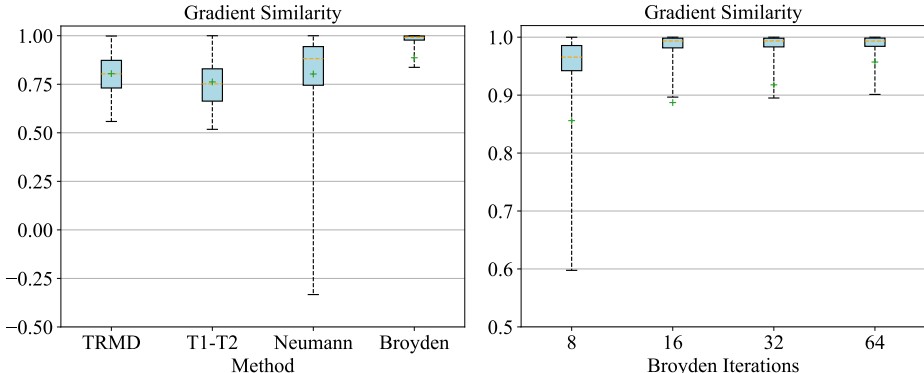

Figure 3: Cosine similarity of hypergradients for different methods (left) and number of iterations (right) on a toy task of Poisson's equation.

close to 1. The other methods also provide a positive approximation that helps to optimize the control variable. However, the stability of Neumann series is not good and in some iterations it provides a negative estimation of hypergradients.

**Fidelity of hypergradients using different number of Broyden iterations.** Since Broyden's method is an iterative algorithm for solving the optimization problem, there is a trade-off between efficiency and performance. Here we compare the fidelity of hypergradients using different numbers of iterations for Broyden's method. We also use box plots for this experiment and the results are in the right part of Figure 3. We observe that as the number of Broyden iterations increases, the hypergradients become more accurate. And the median cosine similarity of hypergradients is more than 0.9 even if we only use 8 iterations which shows the efficiency of Broyden's method. We also find that the increment is minor after 16 iterations for this simple problem. This shows that with the number of iterations increases, Broyden's method converges fast and the rest of the error is dominated by the term caused by inexactness of inner loop optimization in Theorem 1 and Eq. equation 35.

We conduct more ablation studies including the impact of hyperparameters in Broyden's method and PINNs' optimization in Appendix C, the comparison between our BPN and continuous adjoint method with adjoint PDE solved with PINNs in Appendix G, comparison of running time in Appendix G, and provide more visualization results for these PDECO problems in Appendix E.

## 6 CONCLUSIONS

In this paper, we proposed a novel bi-level optimization framework named Bi-level Physics-informed Neural networks (BPN) for solving PDE constrained optimization problems. We used PINNs for solving inner loop optimization and Broyden's method for computing hypergradients. Experiments on multiple PDECO tasks, including complex geometry or non-linear Naiver-Stokes equations, verified the effectiveness of our method. As for potential negative social impact, the interpretability of PINNs is not comparable with traditional numerical solvers, which is left for future work.

## 7 REPRODUCIBILITY STATEMENT

We ensure the reproducibility of our paper from three aspects. (1) Experiment: The implementation of our experiment is described in Sec. 5.1. Ablation study for our experiments is in Sec. 5.5. Further details are in Appendix A and Appendix C. (2) Code: Our code is included in supplementary materials. (3) Theory and Method: A complete proof of the theoretical results described is provided in Appendix B.

## 8 ETHICS STATEMENT

PDE constrained optimization has a wide range of real-world applications in science and engineering including physics, fluids dynamics, heat engineering and aerospace industry, etc. Our BPN is a general framework for PDECO and thus might accelerate the development of these fields. The potential negative impact is that methods based on neural networks like PINNs lack theoretical guarantee and interpretability. Accident investigation becomes more difficult if these unexplainable are deployed in risk-sensitive areas. A possible solution to mitigate this impact is to develop more explainable and robust methods with better theoretical guidance or corner case protection when they are applied to risk-sensitive areas.

## ACKNOWLEDGEMENT

This work was supported by the National Key Research and Development Program of China (2020AAA0106000, 2020AAA0106302, 2021YFB2701000), NSFC Projects (Nos. 62061136001, 62076147, U19B2034, U1811461, U19A2081, 61972224), BNRist (BNR2022RC01006), Tsinghua Institute for Guo Qiang, and the High Performance Computing Center, Tsinghua University. J.Z was also supported by the XPlorer Prize.

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

## A    DETAILS OF PDECO TASKS

In this section, we give a mathematical description of these PDECO tasks in detail.

### A.1    2D POISSON'S EQUATION WITH COMPLEX GEOMETRY (POISSON'S 2D CG)

In this problem, we optimize a two-dimensional Poisson's equation defined on a complex geometry which is a prototype of heat exchanger (Diersch et al., 2011). We solve the equation on a rectangle area $\Omega^r = [-4, 4]^2$ minus four circles $\Omega_i^c = \{(x, y) : (x - x_i)^2 + (y - y_i)^2 \leqslant r_i^2\}$ where $x_i, y_i, r_i$ are $\pm 2.4, \pm 2.4, 0.8$. We define a field variable $u(x, y) \in \mathbb{R}$ on domain $\Omega = \Omega^r \backslash \bigcup_{i=1}^4 \Omega_i^c$. The control source $f(x, y) \in \mathbb{R}$ is distributed on a circle $\chi = \{(x, y) : x^2 + y^2 \leqslant 1.6^2\}$. Our goal is to solve the following optimization problem,

$$
\begin{aligned}
\min_f J &= \frac{1}{|\Omega|} \int_\Omega |u - 1|^2 \mathrm{d}x & (16)\\
s.t. \quad \Delta u &= -f\mathbb{I}\{x \in \chi\}, x \in \Omega \\
u &= 1, x \in \partial\Omega^r \\
u &= 0, x \in \partial\Omega_i^c
\end{aligned}
$$

.

We could visualize the geometry of this problem in Figure 4. The source function is distributed in a circle at the origin with radius of 1.6 (not displayed here).

### A.2    TIME DISTRIBUTED CONTROL OF 2D HEAT EQUATION (HEAT 2D)

In this problem, our goal is to solve an optimal control task of a system governed by the heat equation. The temperature field $u(x, y, t) \in \mathbb{R}$ is defined on a rectangle domain $\Omega = [0, 1]^2$ with time $t \in [0, 2]$ and the control signal is a function $f(t) \in \mathbb{R}$ depends on time but does not depend on spatial coordinates $x$ and $y$. Our goal is to make $u$ close to a target function $\hat{u}(x, y, t)$,

$$
\begin{aligned}
\min_f J &= \frac{1}{2} \int_{\Omega \times [0,2]} |u - \hat{u}|^2 \mathrm{d}x & (17)\\
s.t. \frac{\partial u}{\partial t} - \nu \Delta u &= f, (x, y, t) \in \Omega \times [0, 2] \\
u(x, y, t) &= 0, (x, y, t) \in \partial\Omega \times [0, 2] \\
u(x, y, 0) &= 0, (x, y) \in \Omega
\end{aligned}
$$

The coefficient is $\nu = 0.001$ and the target function is chosen as $\hat{u} = 32x(1 - x)y(1 - y)\sin \pi t$. We choose $f(t) = 0.1$ as the initial guess for the problem. The solution at timestep $t = 2.0$ is shown in Figure 6.

### A.3    TIME DISTRIBUTED CONTROL OF 1D BURGERS EQUATION (BURGERS 1D)

Burgers equation is a nonlinear PDE widely used in various areas like applied mathematics, fluid mechanics, gas dynamics, traffic flow and nonlinear acoustics. Here we use viscous Burgers equations as an example which is a dissipative system. The field variable $u(x, t)$ is defined on $\Omega = [-1, 1]$ and $t \in [0, 1]$. The control variable $f(t)$ depends only on time. We aim to solve the following optimization problem,

$$
\begin{aligned}
\min_f J &= \int_\Omega |u(x, 1) - \hat{u}|^2 \mathrm{d}x & (18)\\
\frac{\partial u}{\partial t} + u \frac{\partial u}{\partial x} - \nu \Delta u &= f, x \in \Omega \\
u(x, t) &= 0, x \in \partial\Omega \\
u(x, 0) &= \sin(\pi x)e^{-2x^2}
\end{aligned}
$$

The diffusion coefficient $\nu = 0.01$ and the target function is chosen as

$$
\hat{u} = e^{-\left(x - \frac{1}{2}\right)^2} - e^{-\left(x + \frac{1}{2}\right)^2}, \tag{19}
$$

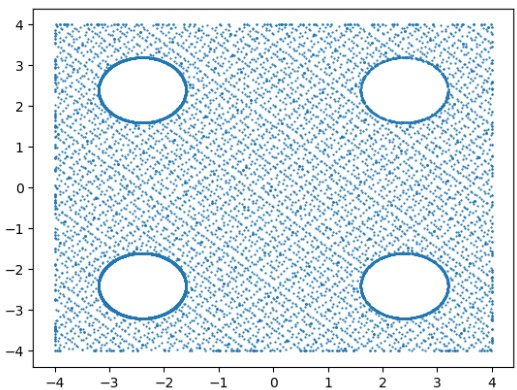

Figure 4: Visualization of geometry shape of Poisson's 2d CG.

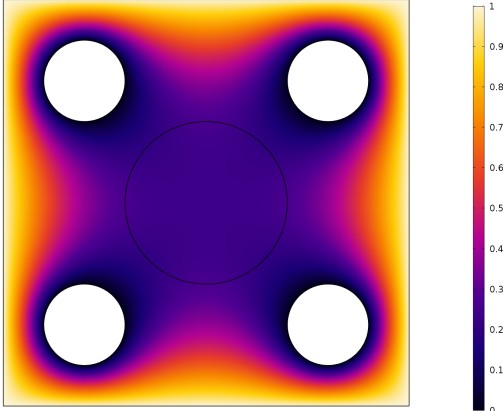

Figure 5: Init guess solution of Poisson's 2d CG. (Simulated using high fidelity FEM)

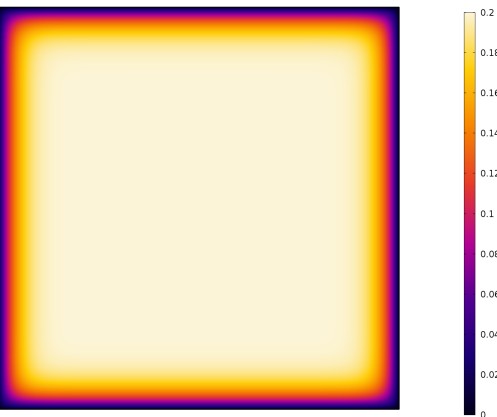

Figure 6: Initial guess solution for heat2d problem at final time $t = 2.0$. The result is simulated by high fidelity FEM.

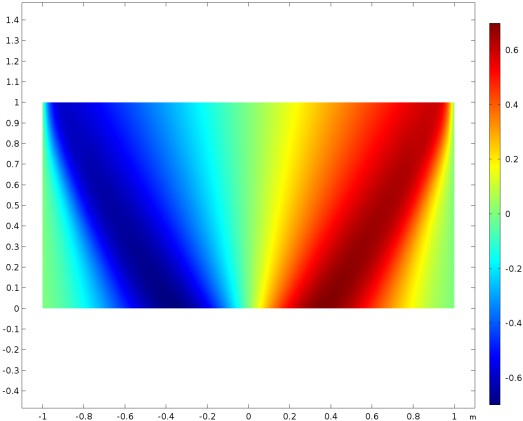

Figure 7: Visualization of the initial guess solution for Burgers 1d problem. The horizontal axis is spatial coordinates $x$ and the vertical axis is time $t$.

which is a symmetric wave. We use $f(t) = 0$ as the initial guess for this problem. The visualization of the initial solution is shown in the Figure 7

### A.4 Inlet flow control for 2d steady Naiver-Stokes Equations (NS 2inlets)

Naiver-Stokes equations are one of the most important equations in fluid mechanics, aerodynamics and applied mathematics which are notoriously difficult to solve due to the high non-linearity. In this problem, we solve NS equations in a pipeline and aim to find the best inlet flow distribution $f(y)$ to make the outlet flow as uniform as possible. The flow velocity field is $\mathbf{u} = (u, v)$ and the pressure field is $p$ and they are defined on a rectangle domain $\Omega = [0, 1.5] \times [0, 1.0]$. We have two inlets and two outlets and several walls for this domain,

$$
\begin{aligned}
\Gamma_1^{\text{in}} &= \{(0, y) : 0 \leqslant y \leqslant 0.5\} \\
\Gamma_2^{\text{in}} &= \{(x, 0) : 0.5 \leqslant x \leqslant 1\} \\
\Gamma_1^{\text{out}} &= \{(1.5, y) : 0 \leqslant y \leqslant 0.5\} \\
\Gamma_2^{\text{out}} &= \{(x, 0.5) : 0.5 \leqslant x \leqslant 1\} \\
\Gamma^w &= \partial\Omega \backslash (\Gamma_1^{\text{in}} \cup \Gamma_2^{\text{in}} \cup \Gamma_1^{\text{out}} \cup \Gamma_2^{\text{out}}).
\end{aligned}
\tag{20}
$$

The whole problem is as follows,

$$
\begin{aligned}
\min_f J &= \int_{\Gamma_1^{\text{out}}} |u - \hat{u}|^2 \mathrm{d}y \\
s.t. (\mathbf{u} \cdot \nabla)\mathbf{u} &= \frac{1}{\text{Re}} \nabla^2 \mathbf{u} - \nabla p \\
\nabla \cdot \mathbf{u} &= 0 \\
\mathbf{u} &= (f(y), 0), (x, y) \in \Gamma_1^{\text{in}} \\
\mathbf{u} &= (0, v_2(x)), (x, y) \in \Gamma_2^{\text{in}} \cup \Gamma_2^{\text{out}} \\
\mathbf{u} &= 0, (x, y) \in \Gamma^w \\
p &= 0, (x, y) \in \Gamma_1^{\text{out}}
\end{aligned}
\tag{21}
$$

The target function is a parabolic function $\hat{u}(x, y) = 4y(1 - y)$ and the velocity field on the second inlet and outlet is $v_2(x) = 18(x - 0.5)(1 - x)$. The Reynold number is set to 100 in this problem. We initialize the solution $f(y)$ the same with the target function, i.e. $f(y) = 4y(1 - y)$. In this case, the outlet velocity and the whole velocity field is shown in the following pictures 8.

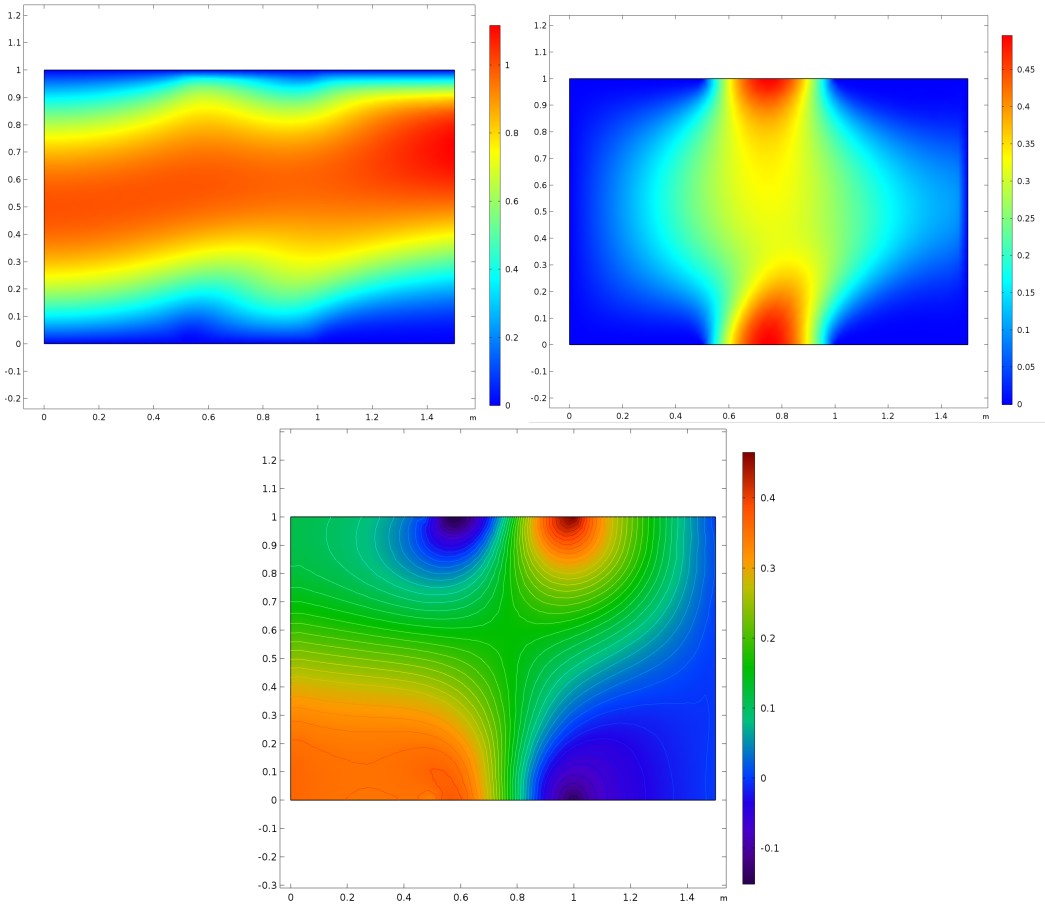

Figure 8: Visualization of the initial guess solution for NS 2inlet. The top two images are velocity fieldd of the X and Y directions. The bottom one is the pressure field.

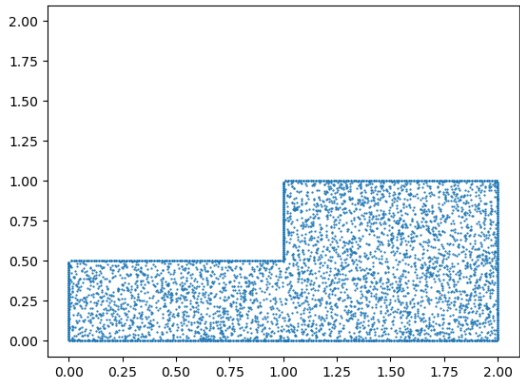

Figure 9: Geometry shape and collocation points for NS 2d backstep problem.

## A.5 DRAG MINIMIZATION OVER AN OBSTACLE OF NS EQUATIONS (NS SHAPE)

This problem is a shape optimization task that is to find the best shape of the obstacle that minimizes the drag forces from the flow. The inlet is the left side of the area and the outlet is the right side of the area,

$$
\begin{aligned}
\Gamma^{\text{in}} &= \{(0, y) : 0 \leqslant y \leqslant 8\} \\
\Gamma^{\text{out}} &= \{(8, y) : 0 \leqslant y \leqslant 8\} \\
\Gamma^{w} &= \partial\Omega \backslash (\Gamma^{\text{in}} \cup \Gamma^{\text{out}})
\end{aligned}
\tag{22}
$$

The flow field is defined on a domain $\Omega = [0, 8]^2 \backslash \Omega^o$ and $\Omega^o$ is the obstacle. The shape of the obstacle is a ellipse parameterized by a parameter $a \in [0.5, 2.5]$. The goal is to minimize the following objective.

$$
\begin{aligned}
\min_{f} J &= \int_{\Omega \backslash \Omega^o} \left(\frac{\partial u}{\partial x}\right)^2 + \left(\frac{\partial u}{\partial y}\right)^2 + \left(\frac{\partial v}{\partial x}\right)^2 + \left(\frac{\partial v}{\partial y}\right)^2 \, \mathrm{d}\mathbf{x} \\
s.t. (\mathbf{u} \cdot \nabla)\mathbf{u} &= \frac{1}{\text{Re}} \nabla^2 \mathbf{u} - \nabla p \\
\nabla \cdot \mathbf{u} &= 0 \\
\mathbf{u} &= \left(\sin(\frac{\pi y}{8}), 0\right), (x, y) \in \Gamma^{\text{in}} \\
\mathbf{u} &= 0, (x, y) \in \Gamma^{w} \\
p &= 0, (x, y) \in \Gamma^{\text{out}}
\end{aligned}
\tag{23}
$$

## A.6 INLET FLOW CONTROL FOR BACKSTEP FLOW (NS BACKSTEP)

The backstep flow is a classic example that might exhibits turbulent flow. In this problem, we also aim to find the best inlet flow distribution $f(y)$ to make the outlet flow symmetric and uniform. The geometry of backstep could be viewed as a union of two rectangles $\Omega = [0, 1] \times [0, 0.5] \cup [1, 2] \times [0, 1]$. The inlet is the left side of the area and the outlet is the right side of the area,

$$
\begin{aligned}
\Gamma^{\text{in}} &= \{(0, y) : 0 \leqslant y \leqslant 0.5\} \\
\Gamma^{\text{out}} &= \{(2, y) : 0 \leqslant y \leqslant 1\} \\
\Gamma^{w} &= \partial\Omega \backslash (\Gamma^{\text{in}} \cup \Gamma^{\text{out}})
\end{aligned}
\tag{24}
$$

We are going to optimize the velocity fields of the outlet,

$$
\begin{aligned}
\min_f J &= \int_{\Gamma_1^{\text{out}}} |u - \hat{u}|^2 \mathrm{d}y && (25) \\
s.t.(\mathbf{u} \cdot \nabla)\mathbf{u} &= \frac{1}{\text{Re}} \nabla^2 \mathbf{u} - \nabla p \\
\nabla \cdot \mathbf{u} &= 0 \\
\mathbf{u} &= (f(y), 0), (x, y) \in \Gamma^{\text{in}} \\
\mathbf{u} &= 0, (x, y) \in \Gamma^w \\
p &= 0, (x, y) \in \Gamma^{\text{out}}
\end{aligned}
$$

The target velocity field is $\hat{u}(x, y) = 3y(1 - y)$ and we initialize the inlet velocity $f(y)$ to be $8y(0.5 - y)$. The Reynold number is also set to 100.

The visualization of the geometry of this problem is shown in Figure 9. A solution of the initial guess is shown in Figure 10.

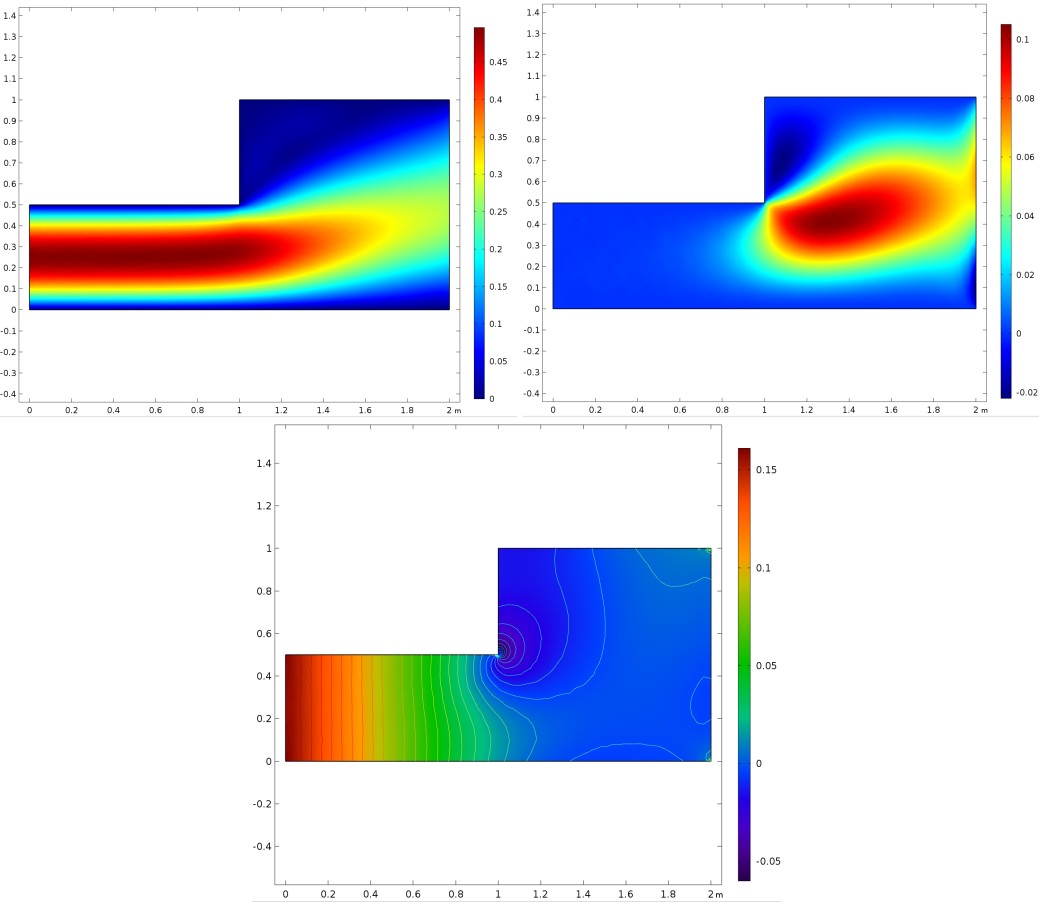

Figure 10: Visualization of the initial guess solution for NS 2d backstep. The top two images are velocity fields of the X and Y directions. The bottom one is the pressure field.

## A.7 DISTRIBUTED FORCE CONTROL OF 2D NAVIER-STOKES EQUATIONS

In previous examples of Naiver-Stokes equations, the control variable is either 1d function (inlet flow) or vector (shape parameters). In this problem, we consider a more challenging setting to control a system governed by NS equation by optimizing distributed force (or source) terms over a domain. The dimension of the control variable (function) in this problem is 2. The geometry and boundary

condition is the same with NS2inlet problem. The flow velocity field is $\boldsymbol{u} = (u, v)$ and the pressure field is $p$ and they are defined on a rectangle domain $\Omega = [0, 1.5] \times [0, 1.0]$. We have two inlets and two outlets and several walls for this domain,

$$
\begin{aligned}
\Gamma_1^{\text{in}} &= \{(0, y) : 0 \leqslant y \leqslant 0.5\} \\
\Gamma_2^{\text{in}} &= \{(x, 0) : 0.5 \leqslant x \leqslant 1\} \\
\Gamma_1^{\text{out}} &= \{(1.5, y) : 0 \leqslant y \leqslant 0.5\} \\
\Gamma_2^{\text{out}} &= \{(x, 0.5) : 0.5 \leqslant x \leqslant 1\} \\
\Gamma^w &= \partial\Omega \backslash (\Gamma_1^{\text{in}} \cup \Gamma_2^{\text{in}} \cup \Gamma_1^{\text{out}} \cup \Gamma_2^{\text{out}}). \\
\Omega^c &= \left\{(x, y) : \sqrt{(x - 0.5)^2 + (y - 0.5)^2} \leqslant 0.2\right\}
\end{aligned}
\tag{26}
$$

The whole problem is as follows,

$$
\begin{aligned}
\min_f J &= \int_{\Gamma_1^{\text{out}}} |u - \hat{u}|^2 \mathrm{d}y \\
s.t. (\boldsymbol{u} \cdot \nabla)\boldsymbol{u} &= \frac{1}{\text{Re}} \nabla^2 \boldsymbol{u} - \nabla p + \boldsymbol{F}(x, y)\mathbb{I}\{\boldsymbol{x} \in \Omega^c\} \\
\nabla \cdot \boldsymbol{u} &= 0 \\
\boldsymbol{u} &= (f(y), 0), (x, y) \in \Gamma_1^{\text{in}} \\
\boldsymbol{u} &= (0, v_2(x)), (x, y) \in \Gamma_2^{\text{in}} \cup \Gamma_2^{\text{out}} \\
\boldsymbol{u} &= 0, (x, y) \in \Gamma^w \\
p &= 0, (x, y) \in \Gamma_1^{\text{out}}
\end{aligned}
\tag{27}
$$

### A.8 POISSION'S 1D TOY PROBLEM FOR EVALUATING HYPERGRADIENTS SIMILARITY.

This is a simple problem with an analytic solution which is a boundary control for one-dimensional Poission's equation. We use this problem for evaluating the fidelity of hypergradients,

$$
\begin{aligned}
\min_{\theta_0, \theta_1} J &= \int_0^1 (y - x^2)^2 \mathrm{d}x \\
s.t. \frac{\mathrm{d}^2 y}{\mathrm{d}x^2} &= 2 \\
y(0) &= \theta_0 \\
y(1) &= \theta_1
\end{aligned}
\tag{28}
$$

The exact solution of this Poission's equation is,

$$
y = x^2 + (\theta_1 - \theta_0 - 1)x + \theta_0.
\tag{29}
$$

And the objective function $J$ is a quadratic form about $\theta_0$ and $\theta_1$,

$$
J = \frac{1}{3}(\theta_0^2 + \theta_1^2 + \theta_1 \theta_0 - 2\theta_1 - \theta_0 + 1).
\tag{30}
$$

The gradient for this function is

$$
\nabla J = \frac{1}{3} \begin{pmatrix} 2\theta_0 + \theta_1 - 1 \\ \theta_0 + 2\theta_1 - 2 \end{pmatrix}.
\tag{31}
$$

Thus we could compare the approximated hypergradients with the real gradients to study the effectiveness of different bi-level optimization strategies. The optimal solution of this problem is

$$
\theta^* = \begin{pmatrix} 0 \\ 1 \end{pmatrix},
$$

and the corresponding objective function $J^* = 0$.

## B PROOF OF THEOREMS

In this section, we prove theorems in the main text.

### B.1 PROOF OF THEOREM 1

**Theorem 1.** *1 If for some $(w', \theta')$, the lower level optimization is solved, i.e. $\frac{\partial \mathcal{E}}{\partial w}|_{(w',\theta')} = 0$ and regularity conditions are satisfied, then there exists a function $w^* = w^*(\theta)$ surrounding $(w', \theta')$ s.t. $\frac{\partial \mathcal{E}}{\partial w}|_{(w^*(\theta'),\theta')} = 0$ and we have:*

$$\frac{\partial w^*}{\partial \theta}\Big|_{\theta'} = -\left[\frac{\partial^2 \mathcal{E}}{\partial w \partial w^\top}\right]^{-1} \cdot \frac{\partial^2 \mathcal{E}}{\partial w \partial \theta^\top}\Big|_{(w^*(\theta'),\theta')} \tag{32}$$

*Proof.* By Cauchy's Implicit Function Theorem, if $f(w', \theta') = 0$ is satisfied and its Jacobian $\frac{\partial f}{\partial w}|_{(w',\theta')}$ is invertible , then there exists an open set $U$ including $\theta'$ and a continuous differentiable function $w^*$ satisfying that $w^*(\theta') = w'$ and $f(w^*(\theta), \theta) = 0, \forall \theta \in U$. Moreover, the partial derivatives could be computed by

$$\frac{\partial w^*}{\partial \theta^\top}(\theta') = -\left(\frac{\partial f}{\partial w}\right)^{-1} \cdot \frac{\partial f}{\partial \theta^\top}|_{(w^*(\theta'),\theta')} \tag{33}$$

Here we take $f(w', \theta') = \frac{\partial \mathcal{E}}{\partial w^\top}(w', \theta')$ and we get the following result

$$\frac{\partial w^*}{\partial \theta} = -\left(\frac{\partial^2 \mathcal{E}}{\partial w \partial w^\top}\right)^{-1} \cdot \frac{\partial^2 \mathcal{E}}{\partial w \partial \theta^\top}|_{(w^*(\theta'),\theta')}. \tag{34}$$

Thus we have proven the theorem. □

### B.2 PROOF OF THEOREM 1

**Theorem 1.** *If assumptions 1 hold and the linear equation 9 is solved with Broyden method, there exists a constant $M > 0$, such that the following inequality holds,*

$$||\mathrm{d}_2 J_t - \mathrm{d}_2 J||_2 \leq (L(1 + \kappa + \frac{\rho M}{\mu^2}) + \frac{\tau M}{\mu})p_t||w||_2 + M\kappa(1 - \frac{1}{m\kappa})^{\frac{m^2}{2}}. \tag{35}$$

*Proof.* First, since we assume the inner loop optimization is a contraction map,

$$\|w_t - w\| = p_t\|w\| \leqslant \|w\| \tag{36}$$

Then for all $t$, the weights $w_t$ are in a compact set $W = \{w' : \|w'\| \leqslant 2\|w\|\}$. Note that $\mathcal{J}$ is continous and differentiable, then there exists a constant $M$ and $\mathcal{J}$ is Lipschitz continous with $M$, i.e. for all $w_t \in W$ we have

$$\|\nabla_1 \mathcal{J}\| \leqslant M. \tag{37}$$

We define the following notations to simplify the proof,

$$\begin{aligned}
\nabla_1 \mathcal{J}_t &= \nabla_1 \mathcal{J}(w_t, \theta) \\
\nabla_1 \mathcal{E}_t &= \nabla_1 \mathcal{E}(w_t, \theta) \\
\nabla_2 \mathcal{E}_t &= \nabla_2 \mathcal{E}(w_t, \theta) \\
z_t &= \nabla_1 \mathcal{J}_t \cdot (\nabla_1^2 \mathcal{E}_t)^{-1} \\
z &= \nabla_1 \mathcal{J} \cdot (\nabla_1^2 \mathcal{E})^{-1}
\end{aligned} \tag{38}$$

and $z_{t,m}$ is the approximation after $m$ iterations of Broyden method. The error of hypergradients are as follows,

$$\begin{aligned}
\|\mathrm{d}\mathcal{J}_t - \mathrm{d}\mathcal{J}\| &= \|\nabla_2 \mathcal{J}_t + z_{t,m}\nabla_{12}\mathcal{E}_t - \nabla_2 \mathcal{J} - z\nabla_{12}\mathcal{E}\| && (39) \\
&\leqslant \|\nabla_2 \mathcal{J}_t - \nabla_2 \mathcal{J}\| + \|z_{t,m}\nabla_{12}\mathcal{E}_t - z\nabla_{12}\mathcal{E}\| && (40) \\
&= \|\nabla_2 \mathcal{J}_t - \nabla_2 \mathcal{J}\| + \|z_{t,m}\nabla_{12}\mathcal{E}_t - z_t\nabla_{12}\mathcal{E} + z_t\nabla_{12}\mathcal{E} - z\nabla_{12}\mathcal{E}\| && (41) \\
&\leqslant \|\nabla_2 \mathcal{J}_t - \nabla_2 \mathcal{J}\| + \|z_t\nabla_{12}\mathcal{E} - z\nabla_{12}\mathcal{E}\| + \|z_{t,m}\nabla_{12}\mathcal{E}_t - z_t\nabla_{12}\mathcal{E}\| && (42)
\end{aligned}$$

Then if we could bound these terms respectively, the total error could be controlled. For the first and second term, it is caused by the inexactness of the inner loop optimization, since $\nabla_2 \mathcal{J}$ is Lipschitz with constant $L$

$$
\begin{aligned}
\|\nabla_2 \mathcal{J}_t - \nabla_2 \mathcal{J}\| &\leqslant L\|w_t - w\| \\
&= Lp_t\|w\|.
\end{aligned}
\tag{43}
$$

And $\nabla_1 \mathcal{E}, \nabla_2 \mathcal{E}$ is Lipschitz continuous with constant $L$,

$$
\begin{aligned}
\|z_t \nabla_{12}\mathcal{E} - z\nabla_{12}\mathcal{E}\| &\leqslant L\|z_t - z\| \\
&= L\|\nabla_1 \mathcal{J}_t \cdot (\nabla_1^2 \mathcal{E}_t)^{-1} - \nabla_1 \mathcal{J} \cdot (\nabla_1^2 \mathcal{E})^{-1}\| \\
&= L\left(\|\nabla_1 \mathcal{J}_t \cdot (\nabla_1^2 \mathcal{E}_t)^{-1} - \nabla_1 \mathcal{J}_t \cdot (\nabla_1^2 \mathcal{E})^{-1}\| + \|\nabla_1 \mathcal{J}_t \cdot (\nabla_1^2 \mathcal{E})^{-1} - \nabla_1 \mathcal{J} \cdot (\nabla_1^2 \mathcal{E})^{-1}\|\right) \\
&\leqslant L\|\nabla_1 \mathcal{J}_t\| \cdot \|(\nabla_1^2 \mathcal{E}_t)^{-1} - (\nabla_1^2 \mathcal{E})^{-1}\| + L\|\nabla_1 \mathcal{J}_t - \nabla_1 \mathcal{J}\| \cdot \|(\nabla_1^2 \mathcal{E})^{-1}\|
\end{aligned}
\tag{44}
$$

And these two terms could be bounded by,

$$
\begin{aligned}
\|(\nabla_1^2 \mathcal{E}_t)^{-1} - (\nabla_1^2 \mathcal{E})^{-1}\| &\leqslant \|\nabla_1^2 \mathcal{E}\| \cdot \|\nabla_1^2 \mathcal{E}_t - \nabla_1^2 \mathcal{E}\| \cdot \|\nabla_1^2 \mathcal{E}\| \\
&\leqslant \frac{\rho}{\mu^2}\|w_t - w\| \\
&\leqslant \frac{\rho}{\mu^2}p_t\|w\|,
\end{aligned}
\tag{45}
$$

$$
\begin{aligned}
\|\nabla_1 \mathcal{J}_t - \nabla_1 \mathcal{J}\| &\leqslant L\|w_t - w\| \\
&\leqslant Lp_t\|w\|.
\end{aligned}
\tag{46}
$$

Thus we have,

$$
\|z_t \nabla_{12}\mathcal{E} - z\nabla_{12}\mathcal{E}\| \leqslant \left(\frac{\rho ML}{\mu^2} + \frac{L^2}{\mu}\right)p_t\|w\|.
\tag{47}
$$

The last term is mixed term caused by both the approximation error when solving the linear system and the inexactness of the inner loop optimization,

$$
\begin{aligned}
\|z_{t,m}\nabla_{12}\mathcal{E}_t - z_t\nabla_{12}\mathcal{E}\| &= \|z_{t,m}\nabla_{12}\mathcal{E}_t - z_t\nabla_{12}\mathcal{E}_t + z_t\nabla_{12}\mathcal{E}_t - z_t\nabla_{12}\mathcal{E}\| \\
&\leqslant \|z_{t,m}\nabla_{12}\mathcal{E}_t - z_t\nabla_{12}\mathcal{E}_t\| + \|z_t\nabla_{12}\mathcal{E}_t - z_t\nabla_{12}\mathcal{E}\| \\
&\leqslant \|\nabla_{12}\mathcal{E}_t\| \cdot \|z_{t,m} - z_t\| + \|z_t\| \cdot \|\nabla_{12}\mathcal{E}_t - \nabla_{12}\mathcal{E}\| \\
&\leqslant L\|z_{t,m} - z_t\| + \|\nabla_1 \mathcal{J}_t\| \cdot \|(\nabla_1^2 \mathcal{E}_t)^{-1}\| \cdot \|\nabla_{12}\mathcal{E}_t - \nabla_{12}\mathcal{E}\| \\
&\leqslant L\|z_{t,m} - z_t\| + \frac{M}{\mu}\|\nabla_{12}\mathcal{E}_t - \nabla_{12}\mathcal{E}\| \\
&\leqslant L\|z_{t,m} - z_t\| + \frac{M\tau}{\mu}\|w_t - w\| \\
&\leqslant L\|z_{t,m} - z_t\| + \frac{M\tau}{\mu}p_t\|w\|
\end{aligned}
$$

since Broyden method enjoys a superlinear convergence rate (Rodomanov & Nesterov, 2021), we have the following convergence bound,

$$
\begin{aligned}
L\|z_{t,m} - z_t\| &\leqslant L\left(1 - \frac{1}{m\kappa}\right)^{\frac{m^2}{2}}\|z_t\| \\
&= L\left(1 - \frac{1}{m\kappa}\right)^{\frac{m^2}{2}}\|\nabla_1 \mathcal{J}_t \cdot (\nabla_1^2 \mathcal{E}_t)^{-1}\| \\
&\leqslant L\left(1 - \frac{1}{m\kappa}\right)^{\frac{m^2}{2}}\|\nabla_1 \mathcal{J}_t\| \cdot \|(\nabla_1^2 \mathcal{E}_t)^{-1}\| \\
&\leqslant L\left(1 - \frac{1}{m\kappa}\right)^{\frac{m^2}{2}}M\frac{1}{\mu} \\
&= M\kappa\left(1 - \frac{1}{m\kappa}\right)^{\frac{m^2}{2}}.
\end{aligned}
\tag{48}
$$

| Iters Rank | 8 | 16 | 32 | 64 |
|---|---|---|---|---|
| 8 | 0.0392 | 0.0390 | 0.0397 | 0.0382 |
| 16 | – | 0.0394 | **0.0379** | 0.0384 |
| 32 | – | – | **0.0379** | 0.0386 |
| 64 | – | – | – | **0.0379** |

Table 3: Ablation Study on Heat2d problem. We report the performance of BPN with different numbers of maximum ranks and iterations for Broyden method.

Thus we sum these error terms together and get,

$$
\begin{aligned}
\|\mathrm{d}\mathcal{J}_t - \mathrm{d}\mathcal{J}\| &\leqslant \|\nabla_2 \mathcal{J}_t - \nabla_2 \mathcal{J}\| + \|z_t \nabla_{12}\mathcal{E} - z\nabla_{12}\mathcal{E}\| + \|z_{t,m}\nabla_{12}\mathcal{E}_t - z_t\nabla_{12}\mathcal{E}\| \\
&\leqslant Lp_t\|w\| + \left(\frac{\rho ML}{\mu^2} + \frac{L^2}{\mu}\right)p_t\|w\| + M\kappa\left(1 - \frac{1}{m\kappa}\right)^{\frac{m^2}{2}} + \frac{M\tau}{\mu}p_t\|w\| \\
&= \left(L\left(1 + \kappa + \frac{\rho M}{\mu^2}\right) + \frac{\tau M}{\mu}\right)p_t\|w\| + M\kappa\left(1 - \frac{1}{m\kappa}\right)^{\frac{m^2}{2}}. \quad (49)
\end{aligned}
$$

Thus we have proven the main theorem for error analysis of the hypergradients computation using Broyden method.

$\square$

## C  SUPPLEMENTARY ABLATION EXPERIMENTS AND DETAILS.

In this section, we provide the results of some ablation studies about our BPN and baselines. We choose Heat 2d and NS backstep as examples for these experiments.

### C.1  PERFORMANCE WITH THE DIFFERENT NUMBER OF BROYDEN ITERATIONS AND MAXIMUM MEMORY STEPS.

In this subsection, we investigate the influence of different iterations and memory limit of Broyden method (Broyden, 1965). We conduct experiments on Heat 2d problem and NS backstep problem. Note that the maximum memory limit should be less than the maximum number of iterations.

The results of Heat 2d equation are shown in Table 3. We see that roughly the performance improves (the objective function decreases) with the increase of maximum memory limit and maximum iterations. In this problem, the rank for approximating the inverse Hessian more than 16 is able to handle this problem. There is no significant gain if we use more than 32 iterations.

The results of NS backstep problem are shown in Table 4. The trend is similar to Heat 2d problem and roughly the performance improves with the increase of Broyden iterations. However, we see that this problem is much more difficult and the performance deteriorates if we only use 8 memory steps even though we use many iterations.

### C.2  PERFORMANCE WITH DIFFERENT NUMBERS OF FINETUNING EPOCHS.

In this subsection, we investigate the influence of different finetune epochs for the inner loop of PINNs, i.e. $N_f$ in Algorithm 1. We choose Heat 2d problem for this experiment and the results are listed in Table 5. We also plot the results in Figure 11. We see that this hyperparameter has a considerable influence on the convergence speed from about 0.2k $\sim$ 0.6k. However, its impact on the final performance is minor within a broad range. We also observe that a small $N_f$ might deteriorate final performance. Thus choosing a moderate $N_f$ is efficient and effective.

| Iters
Rank | 8 | 16 | 32 | 64 |
|---|---|---|---|---|
| 8 | 0.136 | 0.066 | 0.056 | 0.071 |
| 16 | - | 0.048 | **0.037** | **0.037** |
| 32 | - | - | **0.037** | 0.039 |
| 64 | - | - | - | **0.036** |

Table 4: Ablation Study on NS backstep problem. We report the performance of BPN with different numbers of maximum ranks and iterations for Broyden method.

| Finetune epochs | Outer loop iterations | | | | | | | | | |
|---|---|---|---|---|---|---|---|---|---|---|
| | 100 | 200 | 300 | 400 | 500 | 600 | 700 | 800 | 900 | 1000 |
| 32 | 0.0939 | 0.0856 | 0.0721 | 0.0563 | 0.0416 | 0.0384 | 0.0389 | 0.0400 | 0.0398 | 0.0410 |
| 64 | 0.0951 | 0.0845 | 0.0703 | 0.0574 | 0.0444 | 0.0400 | 0.0389 | 0.0384 | 0.0384 | 0.0379 |
| 128 | 0.0870 | 0.0697 | 0.0510 | 0.0430 | 0.0403 | 0.0389 | 0.0396 | 0.0392 | 0.0390 | 0.0386 |
| 256 | 0.0964 | 0.0924 | 0.0862 | 0.0715 | 0.0525 | 0.0428 | 0.0396 | 0.0380 | 0.0380 | 0.0379 |
| 512 | 0.0935 | 0.0836 | 0.0662 | 0.0477 | 0.0418 | 0.0401 | 0.0392 | 0.0387 | 0.0380 | 0.0380 |

Table 5: Results of ablation study on the influence of finetuning epochs of PINNs.

### C.3 PERFORMANCE WITH DIFFERENT NUMBERS OF WARMUP EPOCHS.

In this subsection, we investigate the influence of different finetune epochs for the inner loop of PINNs, i.e. $N_w$ in Algorithm 1. We also choose Heat 2d problem for this experiment. The results with different $N_w$ are shown in Table 6. The observations for this experiment are similar to the ablation study on $N_f$. We see that convergence speed might be different for different optimization schedules of PINNs in the inner loop, but the final performance does not change a lot. In summary, BPN is robust to the choice of these two parameters on Heat 2d problem.

## D BPN FOR SHAPE OPTIMIZATION.

In this subsection, we show that BPN could also be used for a special class of PDECO tasks which are called shape optimization. The main challenge for this is the PDE losses $\mathcal{E}$ and optimization target $\mathcal{J}$ does not directly depend on the shape parameters, but they depend on the collocations points when we sample from the region in practice. Thus calculating gradients for shape parameters is a challenge for shape optimization tasks. Another viewpoint is that the shape parameters contribute to the integral domain but not the objective function. This means that the gradients with respect to $\theta$ will not appear in the computational graph. Specifically, if $\mathcal{J}$ is also an integral over the domain, we have

$$\mathcal{J} = \int_{\Omega(\theta)} J(x)\mathrm{d}x. \tag{50}$$

| Warmup epochs | Outer loop iterations | | | | | | | | | |
|---|---|---|---|---|---|---|---|---|---|---|
| | 100 | 200 | 300 | 400 | 500 | 600 | 700 | 800 | 900 | 1000 |
| 50 | 0.0935 | 0.0836 | 0.0662 | 0.0477 | 0.0418 | 0.0401 | 0.0392 | 0.0387 | 0.0384 | 0.0381 |
| 2000 | 0.0939 | 0.0856 | 0.0721 | 0.0563 | 0.0416 | 0.0384 | 0.0389 | 0.0387 | 0.0386 | 0.0380 |
| 6000 | 0.0951 | 0.0845 | 0.0703 | 0.0574 | 0.0444 | 0.0400 | 0.0389 | 0.0384 | 0.0384 | 0.0379 |
| 10000 | 0.0870 | 0.0697 | 0.0510 | 0.0430 | 0.0403 | 0.0389 | 0.0386 | 0.0382 | 0.0380 | 0.0385 |
| 20000 | 0.0964 | 0.0924 | 0.0862 | 0.0715 | 0.0525 | 0.0428 | 0.0396 | 0.0380 | 0.0381 | 0.0380 |

Table 6: Results of ablation study on the influence of warmup epochs of PINNs.

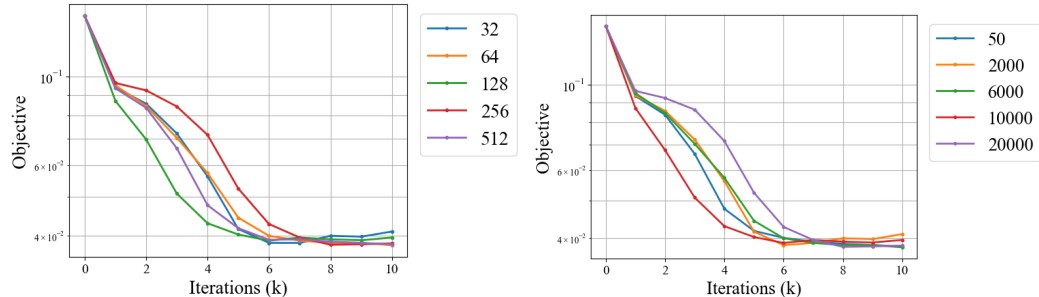

Figure 11: Ablation study on the influence of finetuning steps (left) and warmup epochs (right) on heat 2d problem.

And the PDE losses could be written as (here we assume the control parameters only influence the geometric shape $\Omega(\theta)$),

$$\mathcal{E} = \int_{\Omega(\theta)} |\mathcal{F}(y_w)(x)|^2 \mathrm{d}x + \int_{\partial\Omega(\theta)} |\mathcal{B}(y_w)(x)|^2 \mathrm{d}x. \tag{51}$$

We need to compute the gradients of $\frac{\partial \mathcal{J}}{\partial \theta}, \frac{\partial^2 \mathcal{E}}{\partial w \partial \theta^T}$ which rely on the gradients with respect to $\theta$ that is not contained in the computational graph. Inspired by the idea from traditional shape optimization, we have the following rules,

$$\frac{\mathrm{d}\mathcal{J}}{\mathrm{d}\theta} = \frac{\partial \mathcal{J}}{\partial \theta} + \int_{\partial\Omega(\theta)} J(x) \frac{\partial \boldsymbol{x}}{\partial \theta} \cdot \boldsymbol{n} \mathrm{d}s, \tag{52}$$

where $\boldsymbol{n}$ is normal direction for boundary $\partial\Omega(\theta)$. For the PDE loss, we have

$$\frac{\mathrm{d}\mathcal{E}}{\mathrm{d}\theta} = \int_{\partial\Omega(\theta)} \left( |\mathcal{F}(y_w)|^2 + \frac{\partial}{\partial \boldsymbol{n}} |\mathcal{B}(y_w)(x)|^2 + |\mathcal{B}(y_w)(x)|^2 \kappa \right) \frac{\partial \boldsymbol{x}}{\partial \boldsymbol{n}} \cdot \boldsymbol{n} \mathrm{d}s, \tag{53}$$

where $\kappa$ is the culvature of the boundary shape for two dimensional problems. However, in order to compute the hypergradients, we need to compute the following vector-Hessian product,

$$\begin{aligned} \frac{\partial J}{\partial w^*} \cdot \frac{\partial w^*}{\partial \theta} &= z \cdot \frac{\partial^2 \mathcal{E}}{\partial w \partial \theta^T} \\ &= z \cdot \frac{\partial}{\partial w} \left( \frac{\partial \mathcal{E}}{\partial \theta^T} \right). \end{aligned} \tag{54}$$

We need to compute the directional derivatives for a vector function $\frac{\partial \mathcal{E}}{\partial \theta^T}$. Direct computation of this term requires $n$ times of backpropagation which is extremely inefficient. Here we use a small trick that only requires a constant computational cost by transforming this into a vector-Jacobian product,

$$\frac{\partial J}{\partial w^*} \cdot \frac{\partial w^*}{\partial \theta} = \frac{\partial}{\partial v} \left( z \cdot \left( \frac{\partial}{\partial w} \left( v \cdot \frac{\partial \mathcal{E}}{\partial \theta} \right) \right) \right).$$

We could see that $\frac{\partial}{\partial w} \left( v \cdot \frac{\partial \mathcal{E}}{\partial \theta} \right)$ could be computed using a single vector-Jacobian product and $\frac{\partial}{\partial v} \left( z \cdot \left( \frac{\partial}{\partial w} \left( v \cdot \frac{\partial \mathcal{E}}{\partial \theta} \right) \right) \right)$ could then be computed by two vector-Jacobian products. This trick is easily implemented and efficient in practice.

## E   VISUALIZATION OF OPTIMIZATION RESULTS

In this subsection, we show the results of the PDECO problems. We also compare our optimized result with the reference results solved by high fidelity adjoint solvers based on FEMs. To make sure the evaluation is fair and accurate, we discretize the control variables into meshes. Then we load them into the FEM solver and interpolation the control variable using an interpolation curve or surface. We solve these systems using high-fidelity FEM solvers.

### E.1 POISSION'S 2D EQUATION WITH COMPLEX GEOMETRY.

We show the results of the control variable of Poisson's 2d CG problem solved using our BPN and adjoint method (reference) in Figure 12.

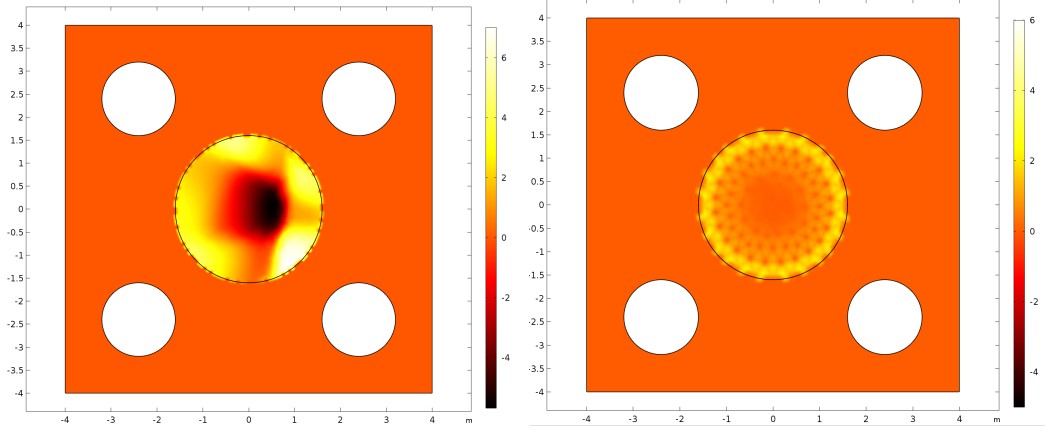

Figure 12: Visualization optimization results of Poisson's 2d CG problem. The left picture is the control variable using BPN and the right picture is the control variable using adjoint method.

And the solution of the PDEs under the optimal control variable is shown in Figure 13.

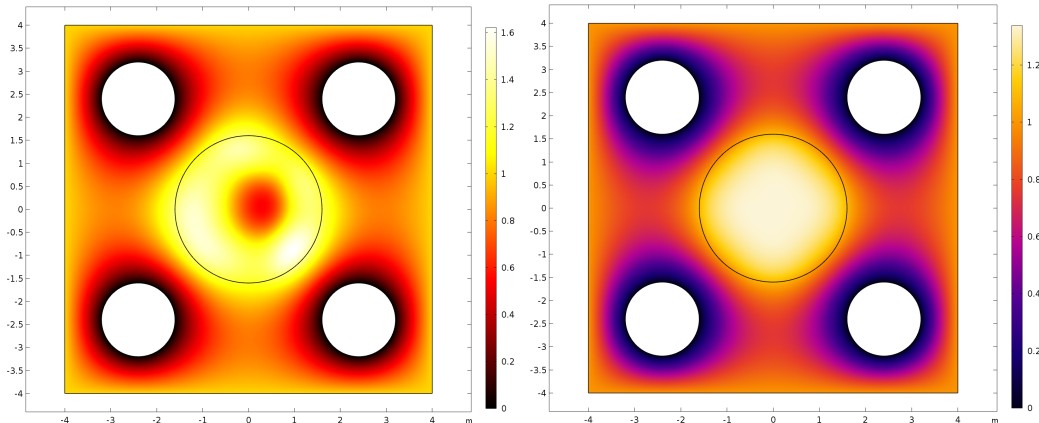

Figure 13: Visualization optimization results of Poisson's 2d CG problem. The left picture is the solution field using BPN and the right picture is the solution field using adjoint method.

We plot the distance between final solution $y$ and the target function in the Figure 14.

We observe that the optimization result using BPN looks different with using adjoint solvers. Although the objective function is nearly the same in this problem, the control variables are different. The results of BPN are not symmetric and is smoother compared with adjoint solvers. A possible reason is that randomness in training PINNs leads to a breaking of symmetry. And the inductive biases of representing control variables using neural networks tend to find smoother solutions.

### E.2 TEMPORAL CONTROL FOR HEAT EQUATIONS.

We show the results of the control variable of Heat 2d problem solved using our BPN and adjoint method (reference) in Figure 15. The solution field of $u$ at time $t = 1.0$ is shown in Figure 16. And the distance to the target function is shown in Figure 17.

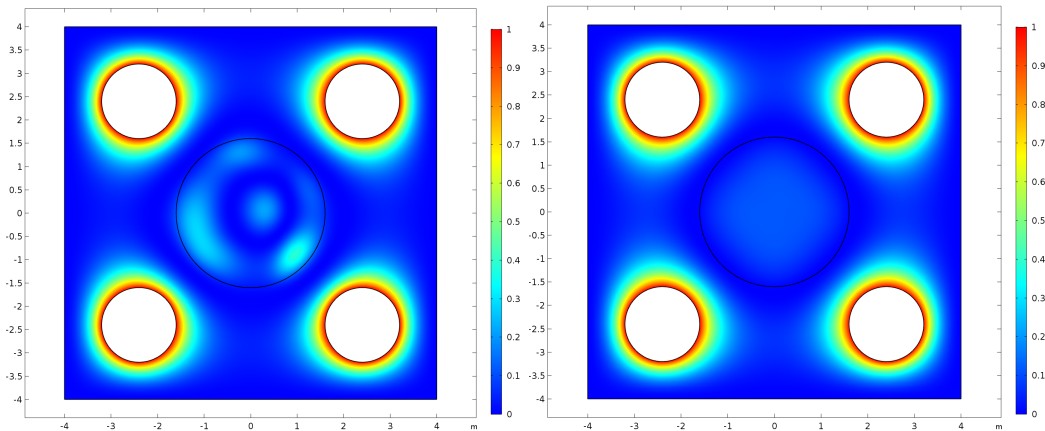

Figure 14: Visualization of error field $(u - \hat{u})^2$ using BPN and adjoint method.

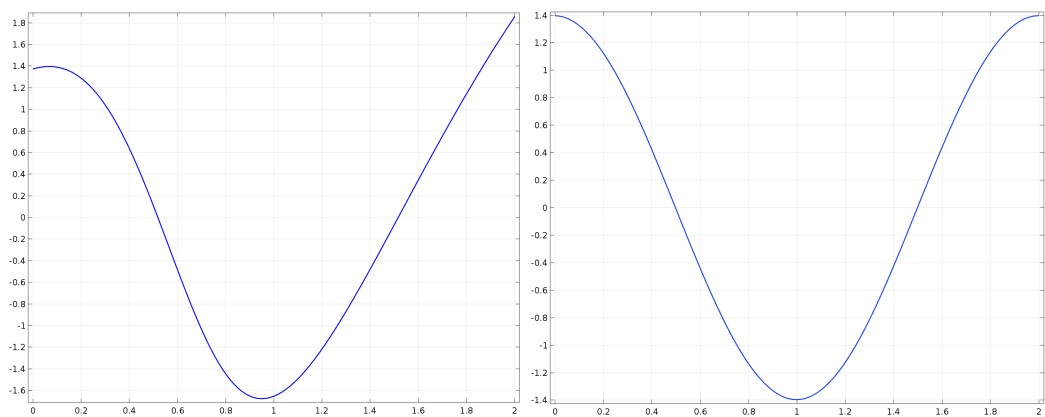

Figure 15: Visualization of control variables.

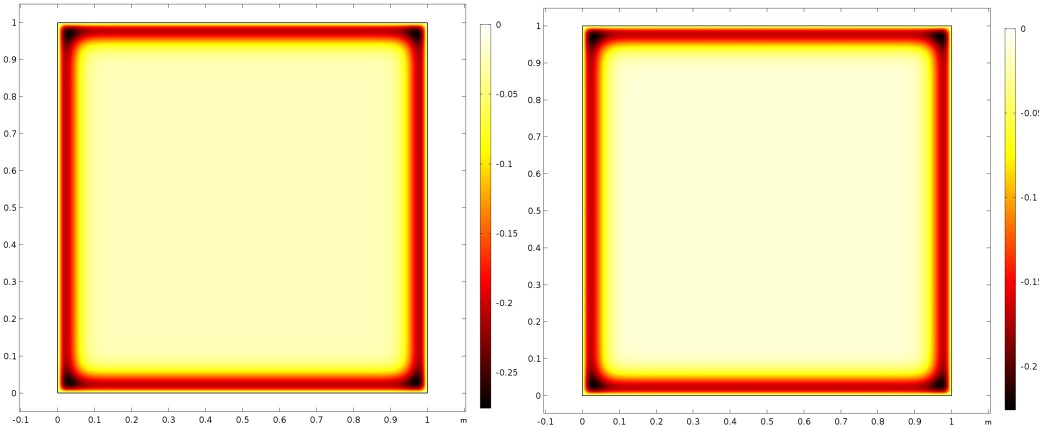

Figure 16: Visualization of solution fields at time $t = 1.0$.

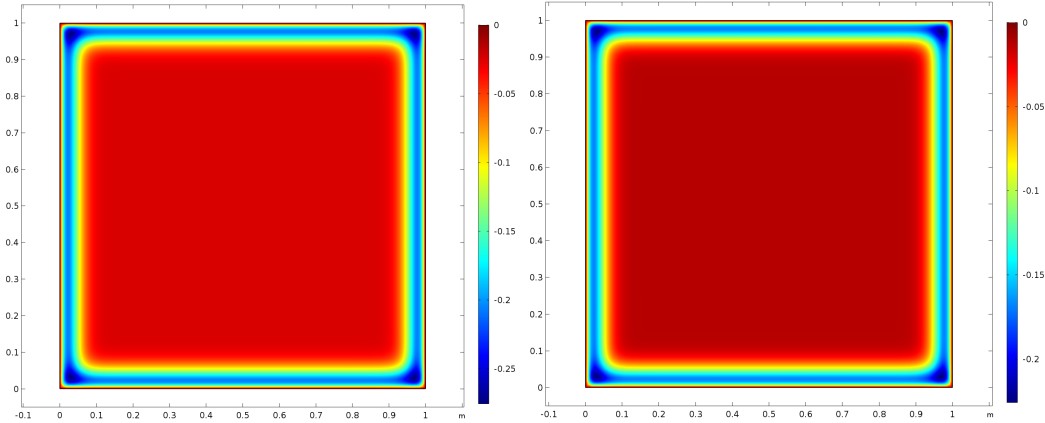

Figure 17: Visualization of error $(u - \hat{u})$ at time $t = 1.0$.

### E.3 NON-LINEAR NAVIER-STOKES EQUATIONS.

Here we only choose the NS backstep problem as an example for visualization of PDECO on Navier-Stokes equations. This problem is more complex due to the non-linearity of Navier-Stokes equations. Both the results of our BPN and adjoint solvers might not be the ground truth solution. We visualize the velocity field $u, v$ of the solution in Figure 18 and Figure 19. The pressure field is shown in 20. We see that both methods allocate a higher velocity to the lower part of the inlet. But the details are different since there are many degrees of freedom for this problem. It is also possible there are many local minima for this problem.

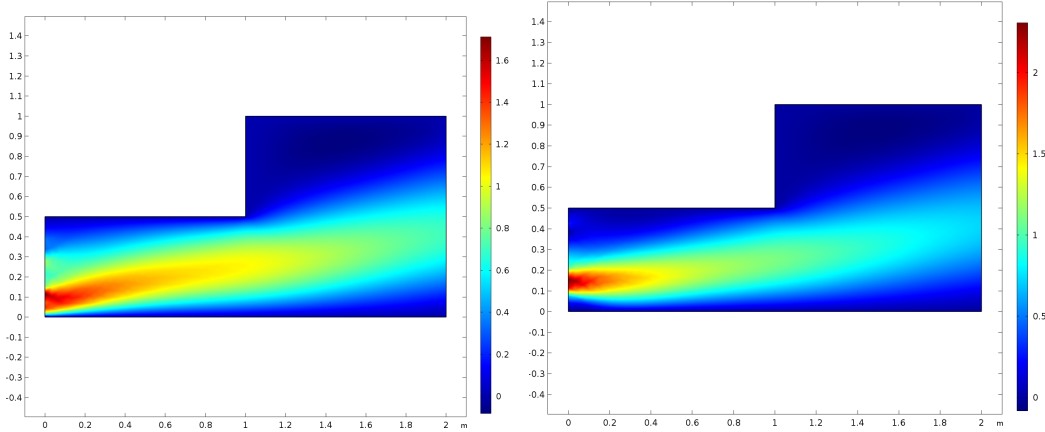

Figure 18: Visualization of solution field $u$.

## F OTHER DETAILS AND HYPERPARAMETERS FOR BASELINES.

In this section, we specify some details and hyperparameters for baselines.

### F.1 HYPERPARAMETERS FOR REGULARIZATION BASED METHODS.

For these methods, we need to adjust the weights of the Lagrangian multipliers after several finetune epochs of PINNs. We choose the finetune steps from $\{1000, 2000, 4000, 6000\}$. For hPINN, the weights are set to 0.001 for initialization and they are amplified by the ratio of 2 after each finetuning stage. The maximum weights are 1e3. For the line search strategy of PINN-LS, the maximum loss threshold is selected from $\{10^{-4}, 10^{-3}, 10^{-2}\}$ for different problems.

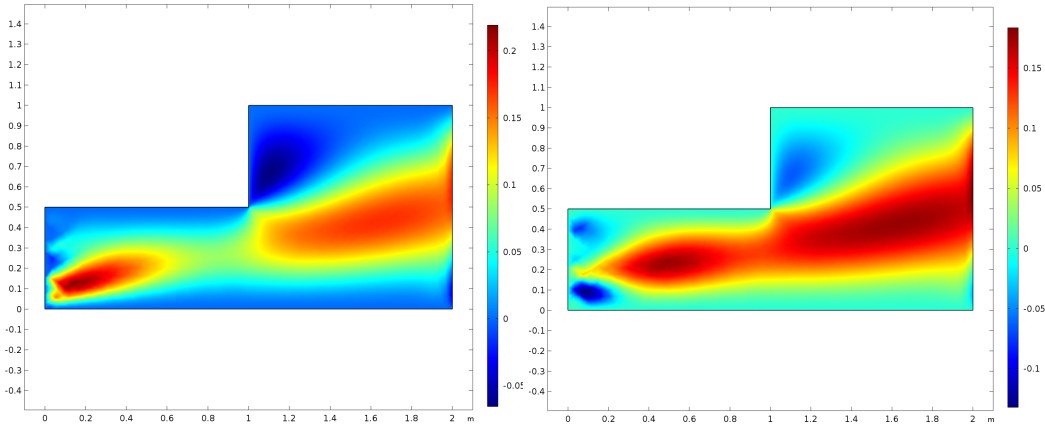

Figure 19: Visualization of solution field $v$.

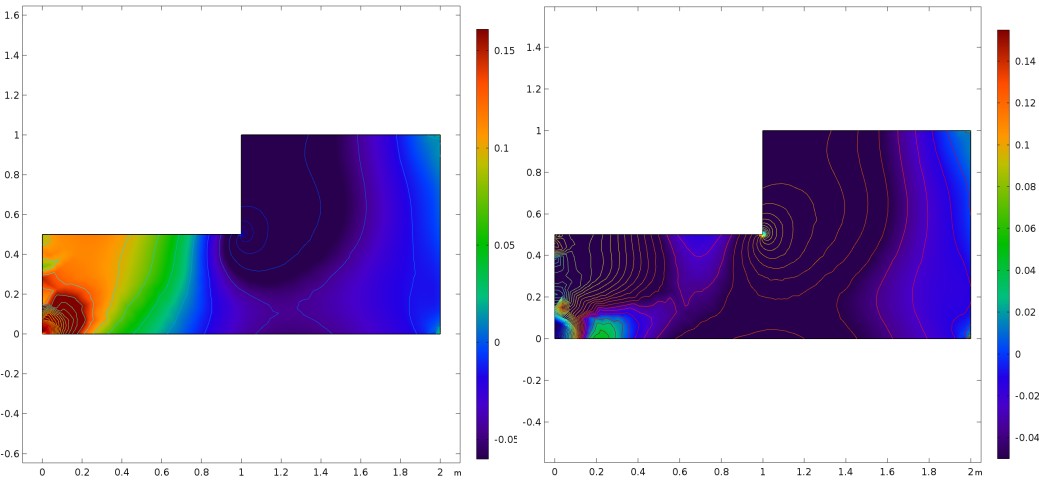

Figure 20: Visualization of solution field $p$.

### F.2 HYPERPARAMETERS FOR PI-DEEPONET.

We use modified MLPs as network architectures for PI-DeepONets proposed by (Wang et al., 2021b;a). The number of layers is selected from $3 \sim 8$ and the layer width is selected from $\{32, 64, 128, 256\}$. We use a Gaussian random field for generating the training functions (Wang et al., 2021a). We generate another 1000 input functions also from the GRF as the test functions. We pretrain the PI-DeepONet until convergence. In the second stage, we replace the PDE constraints with the PI-DeepONet and optimize the control variables using the Adam optimizer with a learning rate of $10^{-3}$.

### F.3 DETAILS OF OTHER BI-LEVEL OPTIMIZATION METHODS.

**TRMD**. It uses a truncated reverse-mode automatic differentiation for computing the hypergradients. If we use the SGD optimizer with $L$ steps as the inner loop update algorithm,

$$w_{i+1} = w_i - \alpha \frac{\partial \mathcal{E}}{\partial w_i}. \tag{55}$$

We need to compute the following Jacobian matrix,

$$\frac{\partial w_{i+1}}{\partial \theta^\top} = \frac{\partial w_i}{\partial \theta^\top} - \alpha \frac{\partial^2 \mathcal{E}}{\partial w_i \partial \theta^\top}. \tag{56}$$

Instead of create an instance of Jacobian matrix, we calculate the following Jacobian-vector product,

$$\frac{\partial \mathcal{J}}{\partial w_L} \cdot \frac{\partial w_{i+1}}{\partial \theta^\top} = \frac{\partial \mathcal{J}}{\partial w_L} \cdot \frac{\partial w_i}{\partial \theta^\top} - \alpha \frac{\partial \mathcal{J}}{\partial w_L} \cdot \frac{\partial^2 \mathcal{E}}{\partial w_i \partial \theta^\top}. \tag{57}$$

We could iteratively calculate it with a truncated history of $m$ steps. The steps is chosen from $\{1, 5, 10, 20\}$ in experiments.

**Neumann Series.** It also approximates the hypergradients based on Implicit Function Differentiation. But the difference is that it uses the following rule to approximate the inverse Hessian matrix,

$$\left( \frac{\partial^2 \mathcal{E}}{\partial w \partial w^\top} \right)^{-1} = \alpha \sum_{k=0}^{\infty} \left( I - \alpha \frac{\partial^2 \mathcal{E}}{\partial w \partial w^\top} \right)^k \tag{58}$$

$$\approx \alpha \sum_{k=0}^{L} \left( I - \alpha \frac{\partial^2 \mathcal{E}}{\partial w \partial w^\top} \right)^k. \tag{59}$$

And the inverse Hessian-Jacobian product is approximated by,

$$\frac{\partial \mathcal{J}}{\partial w} \cdot \left( \frac{\partial^2 \mathcal{E}}{\partial w \partial w^\top} \right)^{-1} \approx \alpha \sum_{k=0}^{L} \frac{\partial \mathcal{J}}{\partial w} \cdot \left( I - \alpha \frac{\partial^2 \mathcal{E}}{\partial w \partial w^\top} \right)^k. \tag{60}$$

The convergence speed is decided by two hyperparameters $\alpha$ and $K$. We choose $\alpha$ from $\{10^{-6}, 10^{-5}, 10^{-4}, 10^{-3}, 10^{-2}\}$ and $K$ from $\{1, 2, 4, 8, 16, 32\}$. In theory, $\alpha$ should satisfy $\alpha < || \left( \frac{\partial^2 \mathcal{E}}{\partial w \partial w^\top} \right) ||_2$. We also experimentally find that large $\alpha$ and large $K$ might lead to divergence depending on the problems. For too small $\alpha$ it might degenerate to the method of $T1 - T2$.

## G SUPPLEMENTARY EXPERIMENTS

In this section, we provide supplementary experimental results to show the effectiveness of our method.

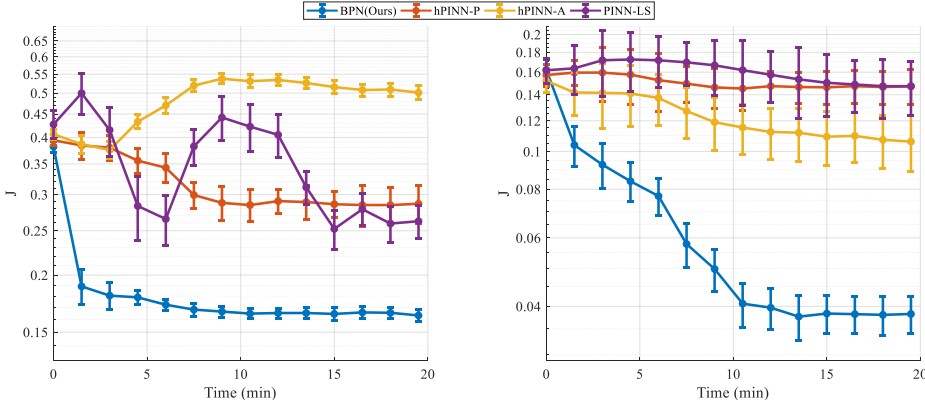

Figure 21: Results of efficiency experiments on Poisson 2d CG (left) and Heat 2d (right) problem. The horizontal axis represents the GPU clock time.

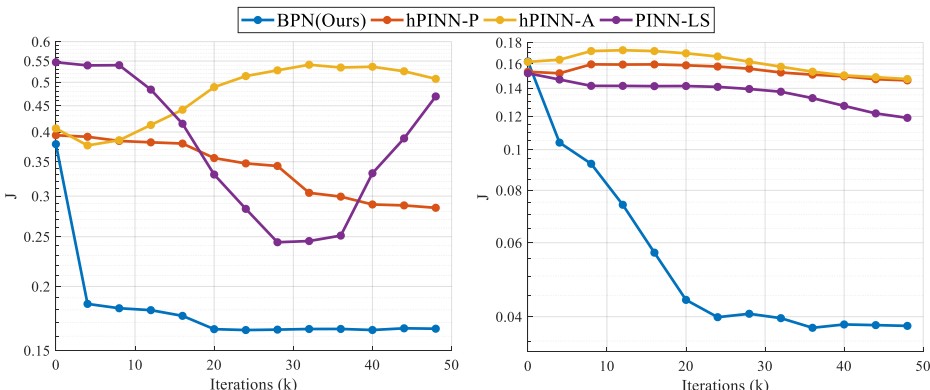

Figure 22: Results of efficiency experiments on Poisson 2d CG (left) and Heat 2d (right) problem. The horizontal axis represents total number of iterations.

### G.1 EFFICIENCY EXPERIMENTS

In this subsection, we conduct efficiency experiments to show that our BPN is more efficient compared with baselines. We measure objective functions varying with both time and number of iterations for baselines on Poisson's 2d CG problem and Heat 2d problem. The results are shown in the following two figures.

### G.2 RUN TIME ANALYSIS AND ERROR BARS

Here we report experimental results measuring error bars and wall clock runtime for these methods on POisson's 2d CG problem and Heat 2d problem.

- We run all experiments with 5 random seeds and compute the mean and std of these results.

- We measure all wall clock time for comparison. For neural networks based methods, we use a single 2080 Ti GPU. The reference FEM are runned on a laptop with 8 threads of CPU (11-th Gen Intel Core i5-11300 H @3.10 GHz ). The mesh size of reference FEM is roughly 500x500 for Poisson's 2d CG and 400x400x20 for Heat 2d. We omit PI-DeepONet for time comparison because it uses a pre-trained operator network.

| Objective:mean/std/time | Poisson's 2d CG | | | Heat 2d | | |
|---|---|---|---|---|---|---|
| | mean | std | time(min) | mean | std | time |
| hPINN-P | 0.368 | 0.025 | 7 | 0.149 | 0.015 | 12 |
| hPINN-A | 0.356 | 0.017 | 14 | 0.147 | 0.023 | 17 |
| hPINN-LS | 0.245 | 0.021 | 9 | 0.107 | 0.017 | 15 |
| PI-DeepONet | 0.376 | 0.035 | – | 0.0512 | 0.0012 | – |
| T1-T2 | 0.412 | 0.031 | 8 | 0.0407 | 0.008 | 14 |
| Neumann | 0.217 | 0.019 | 15 | 0.0750 | 0.023 | 18 |
| TRMD | 0.239 | 0.041 | 28 | 0.0711 | 0.027 | 42 |
| Ours | 0.163 | 0.005 | 18 | 0.0382 | 0.004 | 19 |
| Ref FEM | 0.159 | – | 12 | 0.0378 | – | 15 |

Table 7: Experiments for run time analysis and error bars.

| Iterations | Poisson's 2d CG | | Heat 2d | |
|---|---|---|---|---|
| | Objective (J)(mean/std) | Total Time (min)(mean/std) | Objective (J) | Total Time (min) |
| 50 | 0.232(0.017) | 9 | 0.102(0.019) | 13 |
| 100 | 0.225(0.013) | 11 | 0.106(0.016) | 15 |
| 500 | 0.201(0.015) | 15 | 0.0892(0.017) | 17 |
| 1000 | 0.189(0.016) | 20 | 0.0592(0.013) | 23 |
| 2000 | 0.180(0.008) | 28 | 0.0590(0.007) | 31 |
| Broyden | 0.163(0.005) | 18 | 0.0382(0.004) | 19 |

Table 8: Experimental results for comparsion with computing hypergradients with conjugate gradient method.

### G.3 COMPARISON WITH CONJUGATE GRADIENT METHOD.

We compare the time cost and the performance on Poisson's 2d CG and Heat 2d problems using the conjugate gradient for computing hypergradients. The results are shown in the following table. We use 50 iterations for Broyden's method.

We have two observations from these results. First, the performance using Broyden's method is better than using conjugated gradients. We found the main reason is that the hypergradients are more accurately approximated than using the conjugate gradient. Second, the performance of conjugated gradients improves with the increase in the number of iterations of CG. However, it is not efficient enough to use conjugated descent without preconditioning. But since we cannot store the whole hessian matrix, we are not able to use effective preconditioning techniques like incomplete Cholesky decomposition or Gauss-Seidel preconditioning. In summary, using a quasi-Netwon method like Broyden's method is more efficient for solving such a high dimensional and dense linear equation.

### G.4 COMPARISON WITH ADJOINT SIMULATION WITH PINNs.

Adjoint PDEs for all cases in our main experiments, exist and they can be analytically derived. We also provide supplementary experiments comparing our algorithm with adjoint method simulated by using PINNs. We denote the method computes gradients from adjoint simulation of PDEs using PINNs as PINN-adjoint.

We found that PINN-adjoint is also effective for linear problems. However, by checking residual loss of PDEs and adjoint PDEs, we found that adjoint PDEs are more difficult to solve with PINNs, especially for complex cases like nonlinear Naiver-Stokes equations. In summary, we see that our BPN is still competitive and outperforms PINN-adjoint in average.

| | Poisson's 2d CG | | Heat 2d | | NS backstep | |
|---|---|---|---|---|---|---|
| | Objective | Time | Objective | Time | Objective | Time |
| PINN-adjoint | 0.163 | 24min | 0.0378 | 16min | 0.0720 | >300min |
| BPN (Ours) | 0.160 | 18min | 0.0379 | 19min | 0.0365 | 55min |

Table 9: Experimental results for comparsion with solving adjoint PDE with PINNs.

| Method(NS2inlet force control) | Objective |
|---|---|
| Initial guess | 8.37e-2 |
| hPINN-P | 4.27e-2 |
| hPINN-A | 5.04e-2 |
| PINN-LS | 9.15e-3 |
| TRMD | 6.19e-3 |
| T1-T2 | 5.22e-2 |
| Neumann | 7.20e-3 |
| Ours | **3.24e-3** |
| Reference | 3.17e-3 |

Table 10: Experimental results for NS2inlet force problem.

### G.5 SUPPLEMENTARY EXPERIMENTS ON NS2INLET FORCE PROBLEM.

We have conducted a supplementary experiment on controlling a system governed by NS equation by optimizing a force (source) distributed over a 2d domain. The results are shown in the following Table 10.

### G.6 DETAILS OF EVALUATION PROTOCOL.

Here we specify more details of the evaluation protocol. For the main experiments, we measure the performance using the final prediction when the algorithm stops rather than the best solution it founds. The motivation for this metric is that for practical problems, estimating the ground truth objective is expensive. For efficiency experiments, we measure their performance in each validation epoch to compare the efficiency for these methods.

