# OpenReview forum: "Bi-level Physics-Informed Neural Networks for PDE Constrained Optimization using Broyden's Hypergradients"
_ICLR.cc/2023/Conference — ICLR 2023 poster_

### Official Review · Reviewer_1sqe · 2022-10-19

**Confidence:** 5
**Clarity, Quality, Novelty And Reproducibility:** The paper is clearly written.
**Correctness:** 3
**Technical Novelty And Significance:** 3
**Empirical Novelty And Significance:** Not applicable
**Recommendation:** 6

**Strength And Weaknesses:**

Strengths
- The idea of bi-level optimization is not new, and the main contribution here is applying Broyden’s
method to efficiently approximate the hypergradients in the outer loop. Also, an error bound is derived for the approximation error of the hypergradients, although this is relatively straightforward from existing results.
- Several numerical examples are tested, and the method achieved good accuracy and speed.

Weaknesses
- Poisson and heat equations are simple, and NS is more challenging. However, in all the examples, the control variables are either scalers or 1D function, which are still simple. Problems with control variables of 2D functions should be tested. (Also, it is worth mentioning in Table 1 what type of the control variables is, scalers or 1D functions.)
- The paper only chose deep learning methods as the baseline. Traditional methods should also be used for comparison.
- In Introduction, “this is the first attempt that solves general PDECO problems using a bi-level optimization framework that enjoys scalability and theoretical guarantee” is not correct. There are traditional bi-level optimization methods for general PDECO problems. Probably this is first deep learning based bi-level optimization framework, but I am not sure.
- In Introduction, “We conduct extensive experiments and achieve state-of-the-art results on several challenging PDECO problems with complex geometry or non-linear Naiver-Stokes equations.” “state-of-the-art results” is not correct. As shown in the paper, traditional adjoint method is used as the Reference method and is better than the proposed method in many cases.


**Summary Of The Paper:**

The paper proposed bi-level PINNs for solving PDE-constrained optimization problems. Bi-level PINNs have an inner loop optimization and outer loop optimization. In the inner loop, PINNs are used to solve PDE constraints, and in the outer loop, the control variables are optimized using Broyden’s hypergradients. Numerical experiments are performed to demonstrate the effectiveness of bi-level PINNs, and better results are achieved.

**Summary Of The Review:**

New method is proposed in the paper, and the results look good. However, the tested problems are still simple, and more challenging problems should be tested.

---

> ### Author Response · Authors · 2022-12-08
> **Thank you for your feedback**
>
> Dear Reviewer 1sqe,
>
> Thank you very much for increasing the score and valuable suggestions. We will try our best to improve the paper.
>
> best regards,
>
> Authors

---

### Official Review · Reviewer_yrSS · 2022-10-23

**Confidence:** 4
**Correctness:** 3
**Technical Novelty And Significance:** 3
**Empirical Novelty And Significance:** 3
**Recommendation:** 6

**Clarity, Quality, Novelty And Reproducibility:**

The idea is straightforward, the writing is easy to follow, and the authors also provide open-source codes for reproducibility. However, the novelty seems to be marginal.

**Strength And Weaknesses:**

Strength:
1. The idea of the proposed reformulation is straightforward and extensive experiments verify the effectiveness of this succinct idea.
2. The paper is well-written, and the main idea is easy to follow.
3. The authors conduct a series of ablation studies, especially on the hypergradient approximation, to show the superiority of the recently-proposed Broyden’s method.

Weaknesses:
1. The idea is hard to say exciting, and the contributions can only be summarized as the simple reformulation and the adoption of the recently-proposed IHVPs method.
2. The theoretical analysis mainly inherits from the cited paper [1] with less new findings.

[1] Anton Rodomanov and Yurii Nesterov. Greedy quasi-newton methods with explicit superlinear convergence. SIAM Journal on Optimization, 31(1):785–811, 2021.

**Summary Of The Paper:**

This paper focuses on an emerging and interesting research topic, Physics-informed neural networks (PINNs) for PDE constrained optimization. Different from existing works that use the regularization-based paradigm which is hard to set a proper weights to balance the optimization targets and regularization terms, this paper, for the first time, transforms the constrained optimization problem into a bi-level optimization problem. In addition, by leveraging Broyden’s method to approximate the IHVPs, the authors can more precisely estimate the hypergradient for the outer-loop hypergradient. Extensive experiments on several constrained PDE optimization problems demonstrate the effectiveness of the proposed reformulation.

**Summary Of The Review:**

This paper is generally well-written and the authors focus on an interesting research problem. Although the novelty seems to be marginal, I'd like to accept this paper to encourage more attempts in the area of deep learning for science.

---

### Official Review · Reviewer_RMiZ · 2022-10-24

**Confidence:** 4
**Clarity, Quality, Novelty And Reproducibility:** The paper is well written.
**Correctness:** 4
**Technical Novelty And Significance:** 4
**Empirical Novelty And Significance:** 4
**Recommendation:** 6

**Strength And Weaknesses:**

Strength:
- The bilevel formulation is interesting and novel.
- The experiment is solid compared to other learning baselines.

Weakness:
- No discussion about algorithms used in bi-level optimization. The reviewer also has no idea of popularly used algorithm for bi-level optimization. I think it would be great to see popularly used bi-level algorithms as a baseline. PINN doesn't use large network. Inverse a Hessian will not be so expansive in my mind.
- No baseline using traditional methods provided. I still doesn't know the benefit of using a NN for PDE in low dimension.
- number of parameter and time of computation is not provided in this version.

**Summary Of The Paper:**

This paper formulates the PDE-constrained optimization problem as a bi-level optimization problem. The gradient of the bilevel problem (using implicit function theorem) involves an inversion of the Hessian. This paper use Broyden's hyper gradient to approximate the inverse of the Hessian.

**Summary Of The Review:**

Overall this is a novel paper definitely.  But this paper lacks literature review of bi-level optimization and PDE constrained optimization at this point.

---

> ### Author Response · Authors · 2022-12-08
> **Thank you for your feedback**
>
> Dear Reviewer RMiZ,
>
> Thank you very much for increasing the score and valuable suggestions. We will try our best to improve the paper.
>
> best regards,
>
> Authors

---

### Official Review · Reviewer_isLd · 2022-10-24

**Confidence:** 4
**Correctness:** 3
**Technical Novelty And Significance:** 3
**Empirical Novelty And Significance:** 2
**Recommendation:** 6

**Clarity, Quality, Novelty And Reproducibility:**

I have a few questions regarding the clarity.
- What is the evaluation metric to measure the performance in Table 1 and 2? This is not clear from section 5.1. Do you have any alternative for evaluation in case that the FEM of PDE is not avaiable?
- The z in eq 11, is it z_i?
- What is t in Assumption 1, and Theorem 1? What is d_2 J? Why the w in the last last point of the assumption 1 and when it holds for PINNs with p_t → 0?

**Strength And Weaknesses:**

The proposed method uses an interesting way to compute the hyper-gradients which is not so standard literature (i.e. to solve eq (8) by eq (9)), but this does not seem to be new. The strength of this paper is to show that the proposed method works well on a wide range of constrained PDE.

**Summary Of The Paper:**

This paper proposes a bi-level optimization method to solve constrained PDE. The key is to compute a hyper-gradient with a high-efficiency and accuracy. The paper has some unclear notations, making it hard for me to understand the main results.

**Summary Of The Review:**

REVISED: I now understand that the evaluation metric is used to show how well how control variable is found by the proposed method. The main results (table 1) show that it work better state-of-the art methods. Therefore I think the paper could be considered to be published. However, I still think that the evaluation on how well the pde is solved for a given control variable, should also be evaluated (maybe in a future work). For this reason, I raise my score to 6 .

---

### Decision · Program_Chairs · 2023-01-20

**Decision:**

Accept: poster

**Justification For Why Not Higher Score:**

Although finally everyone assigned a positive score, none of the reviewers was highly enthusiastic. Our conclusion was that this is a solid piece of work that should be published at our conference, but at the same time, this paper will probably not have the role of a game changer in this field.

**Justification For Why Not Lower Score:**

In the end, all reviewers (and myself) agreed that the positive aspects dominate the weaknesses.

**Metareview: Summary, Strengths And Weaknesses:**

In the end, all reviewers converged to  a positive score. They mentioned some weaknesses, such as (partly) unclear novelty and (sometimes) not so clear experimental comparisons (for instance, with classical adjoint methods).
However, over-all the positive aspects were considered more important:
- the main idea is interesting, clearly motivated and well-described
- the proposed method is applicable and seems to works very well on a wide range of PDEs
- the experimental evaluation is (mostly) convincing.
I share this (slightly) positive impression of this paper, and therefore I vote for acceptance.

**Note From Pc:**

if the above contains the word "oral" or "spotlight" please see: "oral" presentation means -> notable-top-5% and "spotlight" means -> notable-top-25%. As stated in our emails, we are disassociating presentation type from AC recommendations

**Summary Of Ac-Reviewer Meeting:**

Initially, I thought that this was a borderline paper, but during the discussions all reviewers quickly converged to a positive over-all view. Thus, an additional meeting was not necessary in the end.